# Dynamic BMP signaling polarized by Toll patterns the dorsoventral axis in a hemimetabolous insect

Lena Sachs[1†], Yen-Ta Chen[1†], Axel Drechsler[1,2], Jeremy A Lynch[1,3], Kristen A Panfilio[1], Michael Lässig[4], Johannes Berg[4], Siegfried Roth[1*]

[1]Institute for Developmental Biology, University of Cologne, Köln, Germany; [2]Bundesministerium für Umwelt, Naturschutz, Bau und Reaktorsicherheit, Bonn, Germany; [3]Department of Biological Sciences, University of Illinois at Chicago, Chicago, United States; [4]Institute for Theoretical Physics, University of Cologne, Cologne, Germany

**Abstract** Toll-dependent patterning of the dorsoventral axis in *Drosophila* represents one of the best understood gene regulatory networks. However, its evolutionary origin has remained elusive. Outside the insects Toll is not known for a patterning function, but rather for a role in pathogen defense. Here, we show that in the milkweed bug *Oncopeltus fasciatus*, whose lineage split from *Drosophila's* more than 350 million years ago, Toll is only required to polarize a dynamic BMP signaling network. A theoretical model reveals that this network has self-regulatory properties and that shallow Toll signaling gradients are sufficient to initiate axis formation. Such gradients can account for the experimentally observed twinning of insect embryos upon egg fragmentation and might have evolved from a state of uniform Toll activity associated with protecting insect eggs against pathogens.

*For correspondence: siegfried.
roth@uni-koeln.de

†These authors contributed equally to this work

## Introduction

In the fly *Drosophila melanogaster*, the Toll pathway has essential functions both for innate immunity and for dorsoventral (DV) axis formation (*Leulier and Lemaitre, 2008*; *Stein and Stevens, 2014*). While Toll's immune function is broadly conserved in animals ranging from hydra to humans, its role in axis formation appears to be an evolutionary novelty of insects (*Leulier and Lemaitre, 2008*; *Franzenburg et al., 2012*; *Gilmore and Wolenski, 2012*). Other animals do not employ Toll but rather use BMP signaling to establish their DV axis (*De Robertis, 2008*). BMP signaling also plays a crucial, but spatially restricted role in *Drosophila* DV patterning (*O'Connor et al., 2006*). This suggests that Toll signaling was recruited into an ancestral BMP-based patterning network during evolution of the insect lineage.

So far molecular studies of DV patterning in insects have been largely restricted to the most speciose supraorder, Holometabola, the insects with complete metamorphosis (*Lynch and Roth, 2011*). However, already within the Holometabola, a clear evolutionary trend was observed: the more basally branching lineages show an increased reliance on BMP signaling while the importance of Toll signaling is reduced (*Figure 1*). In *Drosophila*, both the polarity and pattern of the DV axis depend on a stable long range gradient of Toll signaling that promotes the graded nuclear uptake of the NF-κB transcription factor Dorsal (*Reeves and Stathopoulos, 2009*). NF-κB/Dorsal acts in a concentration-dependent manner to activate or repress genes required for DV cell fate specification (*Figure 1*). The ventral cell fates of the mesoderm, mesectoderm and neuroectoderm directly depend on NF-κB/Dorsal target genes. The dorsal cell fates (non-neurogenic ectoderm and extraembryonic

**eLife digest** How an animal develops from a fertilized egg has fascinated scientists for decades. As such, much effort has gone into answering the related question: what makes the belly (or underside) of an animal develop differently from its back?

Like almost all other biological processes, the development of an embryo is controlled by interactions between different molecules within cells and tissues. Some of these molecules promote the activity of others; some have the opposite effect; and together these molecules and their interactions form 'signaling networks'. One such network, which involves a protein called BMP, is needed to establish the belly-to-back axis of nearly all animals. However, insects are a unique exception. Most insects (including flies, beetles and wasps) use a different signaling network to control their development from their belly to their back, one that involves a protein called Toll instead. This is unexpected because, in other animals, Toll proteins are best known for their role in the immune system; and it remains unclear how Toll signaling came to be involved in insect development.

Now, Sachs, Chen et al. have studied an insect—called the milkweed bug—that is unlike most insects in that it does not have a larval stage (i.e., a maggot or a caterpillar) in its life-cycle. This characteristic makes the milkweed bug more similar to the ancestor of all insects, and thus makes it an excellent model to study how the Toll protein took over from BMP in insect development.

First, Sachs, Chen et al. experimentally reduced BMP signaling in milkweed bug embryos. This caused the embryos to develop features all around their bodies that are normally only associated with the animal's underside. In other insects, the development of these so-called 'ventral' features is typically controlled by Toll signaling; but in the milkweed bug this activity instead depends on a protein called Sog. Indeed, when Sachs, Chen et al. experimentally reduced both BMP and Toll signaling, the effect was the same as having reduced only BMP signaling, implying that Toll is not needed. Instead, Toll increased the level of the Sog protein up to a particular threshold. Above this threshold, Sog and BMP control each other to set out the animal's body plan. As insects evolved, it seems likely that Toll transitioned from being a trigger of BMP signaling to an important controller of insect development in its own right. But why was Toll put in the egg in the first place? It is possible that Toll was required to protect the eggs of early insects from attack by bacteria and fungi. Future work will now test this assumption and aim to explain how and why the Toll protein changed its role—from immunity to development—during evolution.

amnioserosa) are determined in a more indirect way by Toll signaling restricting and polarizing an opposing BMP signaling gradient (*O'Connor et al., 2006*; *Hong et al., 2008*). The gene regulatory network (GNR) controlled by NF-κB/Dorsal has been extensively characterized. It encompasses 60–70 target genes which fall into six classes according to their enhancer structure (*Hong et al., 2008*). The sensitivity of these enhancers to different NF-κB/Dorsal concentrations is fine-tuned by ubiquitously distributed activators and repressors (*Garcia and Stathopoulos, 2011*; *Ozdemir et al., 2014*).

All major components of the BMP signaling network are controlled by NF-κB/Dorsal: one of the BMP ligands, the BMP2/4 homolog *decapentaplegic* (*dpp*), and the extracellular protease *tolloid* (*tld*) are repressed by NF-κB/Dorsal and thus confined to the dorsal side of the embryo while an extracellular BMP inhibitor, the *chordin* homolog *short gastrulation* (*sog*), and a transcriptional repressor of BMP target genes (*brinker, brk*) are activated by NF-κB/Dorsal at the ventral side (*Jazwinska et al., 1999*; *O'Connor et al., 2006*; *Hong et al., 2008*; *Rushlow and Shvartsman, 2012*). The ventral-to-dorsal transport of Sog and Sog-BMP complexes and their dorsal cleavage by Tld leads to a BMP signaling gradient with peak levels at the dorsal side. Thus, Toll signaling via NF-κB/Dorsal not only provides precise spatial information for the ventral half of the axis, but indirectly also determines the patterning of the dorsal half. An independent maternal input at the dorsal side of *Drosophila* embryos has been discussed, but it apparently plays only a minor role (*Araujo and Bier, 2000*).

In contrast to *Drosophila*, Toll signaling in the beetle *Tribolium castaneum* is highly dynamic due to positive and negative feedback of Toll pathway components (*Nunes da Fonseca et al., 2008*). These dynamics lead to a temporally shifting NF-κB/Dorsal gradient which refines and disappears before the major DV patterning genes have established stable expression domains. This suggests that

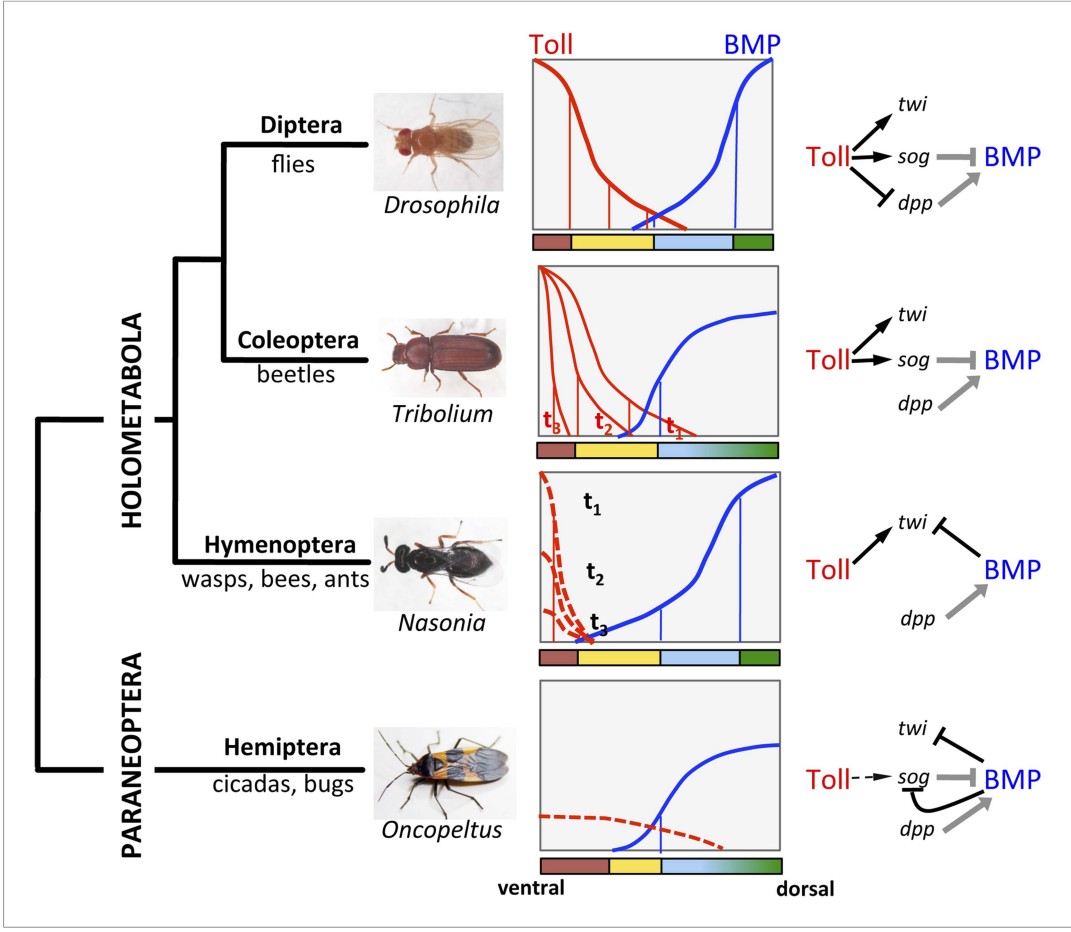

**Figure 1**. The evolution of Toll's role in dorsoventral (DV) patterning in insects. In holometabolous insects Toll signaling is activated by ventral eggshell cues and forms an activity gradient (red) that is essential at the very least for specifying the ventral-most cells on the DV axis, giving rise to the mesoderm (brown), by activating the gene *twist* (*twi*) (black arrow). In the fly *Drosophila* Toll signaling not only determines the mesoderm, but also the neuroectoderm (yellow) and restricts BMP signaling to the dorsal side through several parallel mechanisms, including the activation of the BMP inhibitor *short gastrulation* (*sog*) (black arrow) and repression of the major BMP ligand *decapentaplegic* (*dpp*) (black T-bar) (**Hong et al., 2008**; **Reeves and Stathopoulos, 2009**). On the dorsal side a BMP gradient (blue) is established (gray arrow and T-bar indicate BMP ligand production and inhibition, respectively) that specifies non-neurogenic ectoderm (blue) and extraembryonic tissue (green) (**O'Connor et al., 2006**). Toll signaling is dynamic in *Tribolium* and polarizes BMP signaling only by activating sog (**Nunes da Fonseca et al., 2008**). BMP signaling in turn has an increased role in ectodermal patterning compared to flies (**van der Zee et al., 2006**). In contrast to both *Drosophila* and *Tribolium* Toll signaling in the wasp *Nasonia* appears to be restricted to a narrow ventral region where it is only transiently active. Here, Toll signaling is required to induce mesodermal and mesectodermal fates. But the size of the mesodermal region as well as the fate and position of all other regions along the DV axis are determined by a BMP signaling gradient emanating from the dorsal side by an unknown (Toll-independent) mechanism (black T-bar indicates repression of *twi*) (**Özüak et al., 2014a, 2014b**). Thus, in the holometabolous insects BMP signaling gets increasingly more important towards basally branching groups, while Toll's role is diminished, but remains essential for ventral-most cell fates. Here we provide evidence that the bug *Oncopeltus*, representing the Hemiptera within the sister group of Holometabola (Paraneoptera), uses Toll signaling only as spatial cue (dashed black arrow) to polarize a dynamic BMP signaling network that establishes a gradient responsible for patterning the cell fates along the DV axis. The key regulatory element of this network is the transcriptional repression of *sog* by BMP signaling. A reaction-diffusion model which incorporates this regulatory element shows that the formation of stable BMP gradients requires only weakly polarized Toll signaling (**Box 1**).

NF-κB/Dorsal concentration thresholds play a less direct role in specifying these domains (**Chen et al., 2000**). In addition, NF-κB/Dorsal does not act as a repressor of BMP signaling components or as an activator of *brk* (**Nunes da Fonseca et al., 2010**). Consequently, the establishment of the BMP

gradient entirely relies on ventral (NF-κB/Dorsal dependent) activation of *sog*. The BMP gradient in turn is required for all the polarity of the ectoderm (*van der Zee et al., 2006*). Thus, in *Tribolium* the direct role of Toll signaling is largely restricted to mesoderm and mesectoderm (*Figure 1*).

Finally, in the wasp *Nasonia*, representing the basal-most branch (Hymenoptera) of the Holometabola, Toll signaling appears to be active only in a narrow domain along the ventral midline, where it is required to induce ventral-most cell fates (*Özüak et al., 2014b*). However, the borders of the ventrally expressed genes are not defined by thresholds of Toll signaling, but rather by repressive BMP signaling. Thus, in *Nasonia* BMP signaling specifies gene expression domains along the entire DV axis (*Figure 1*). In this respect the DV system in *Nasonia* is similar to the ancestral type of DV axis formation in bilaterian animals. However, a closer look at the mechanisms of gradient formation reveals that the *Nasonia* system is highly derived even when compared to *Drosophila*. Functional studies show that the BMP gradient of *Nasonia* is established from a maternal source along the dorsal midline independent from ventral Toll signaling (*Özüak et al., 2014b*). Indeed, the *Nasonia* genome lacks a *sog* homolog and no ventrally expressed BMP inhibitor was identified (*Özüak et al., 2014a*). The establishment of BMP signaling gradients by an opposing inhibitor gradient of Chordin/Sog is however, one of the most conserved aspects of DV axis formation in Bilateria and is even preserved in flies (*De Robertis, 2008*). Moreover, given the fact that *Nasonia* also uses Toll for mesoderm/mesectoderm induction, it establishes its DV axis in a bipolar manner employing independent signaling sources along the ventral and dorsal midline of the egg. Bipolar DV axis formation has so far not been described in any other system.

Despite all the variability found so far in Holometabola there are two common themes. (1) In more basal lineages BMP signaling is responsible for functions that are performed by Toll signaling in more derived lineages. (2) The ventral-most regions of the DV axis, giving rise to the mesoderm and mesectoderm, remain strictly dependent on Toll signaling. By studying a representative of insects with incomplete metamorphosis (Hemimetabola) we asked whether this situation is characteristic for all insects or whether a further reduction of the DV patterning function of Toll can be observed, allowing us to analyze how it originated.

To this end we investigated DV patterning in the milkweed bug, *Oncopeltus fasciatus*, representing the order Hemiptera, within the sister group (Paraneoptera) to the Holometabola (*Liu and Kaufman, 2009*). We provide evidence that in *Oncopeltus* Toll is indeed no longer essential for mesoderm formation since repression of BMP signaling suffices to induce mesoderm. Like in other systems inhibition of BMP signaling is accomplished by *sog*. However, the transcriptional regulation of *sog* in *Oncopeltus* is more dynamic than in the other well-studied systems. It combines uniform Toll-independent activation with ventral enhancement by Toll and repression by BMP. We build a theoretical model based on the experimental findings and show that the BMP/sog pathway in *Oncopeltus* exhibits self-organized patterning (*Box 1*). Specifically, the interplay of BMP-dependent *sog* repression and Sog-dependent BMP transport generates a Turing instability (*Turing, 1952*). Toll's role in this system seems to be reduced to providing a trigger that enhances Sog activity above a certain threshold to initiate the patterning process. However, this patterning mechanism differs from the well-studied activator-inhibitor models (*Gierer and Meinhardt, 1972*); while *sog* is inhibited by BMP, there is no activator in our model.

## Results

To mark different DV regions of *Oncopeltus* blastoderm embryos we chose *twist* (*twi*), a ventrally expressed marker for the mesoderm (*Thisse et al., 1988*) (*Figure 2A* lateral view), *single minded* (*sim*), a mesectodermal marker (*Thomas et al., 1988*) expressed in lateral stripes bordering the mesoderm and in a ventral-anterior domain (*Figure 2B* lateral view), and *short gastrulation* (*sog*), the insect homolog of the BMP antagonist Chordin (*Francois et al., 1994*), which is expressed in a ventral domain slightly broader than that of *twi* (*Figure 2C* lateral view).

None of the known dorsally expressed genes from the Holometabola showed specific dorsal expression in early *Oncopeltus* embryos. This includes both the BMP signaling components (*O'Connor et al., 2006*) *decapentaplegic* (*dpp*), *glass bottom boat* (*gbb*), *tolloid* (*tld*) and *twisted gastrulation* (*tsg*) (*Figure 2—figure supplement 1*) as well as target genes potentially activated by BMP signaling like *zerknüllt*, *pannier*, *dorsocross* and *iroquois* (*Panfilio et al., 2006*; *Nunes da Fonseca et al., 2010*; *Buchta et al., 2013*) (data not shown). In the absence of dorsal marker genes we monitored the distribution of phosphorylated Mad (pMAD), the activated form of the transcription factor downstream of BMP signaling (*Dorfman and Shilo, 2001*; *van der Zee et al., 2006*). In early

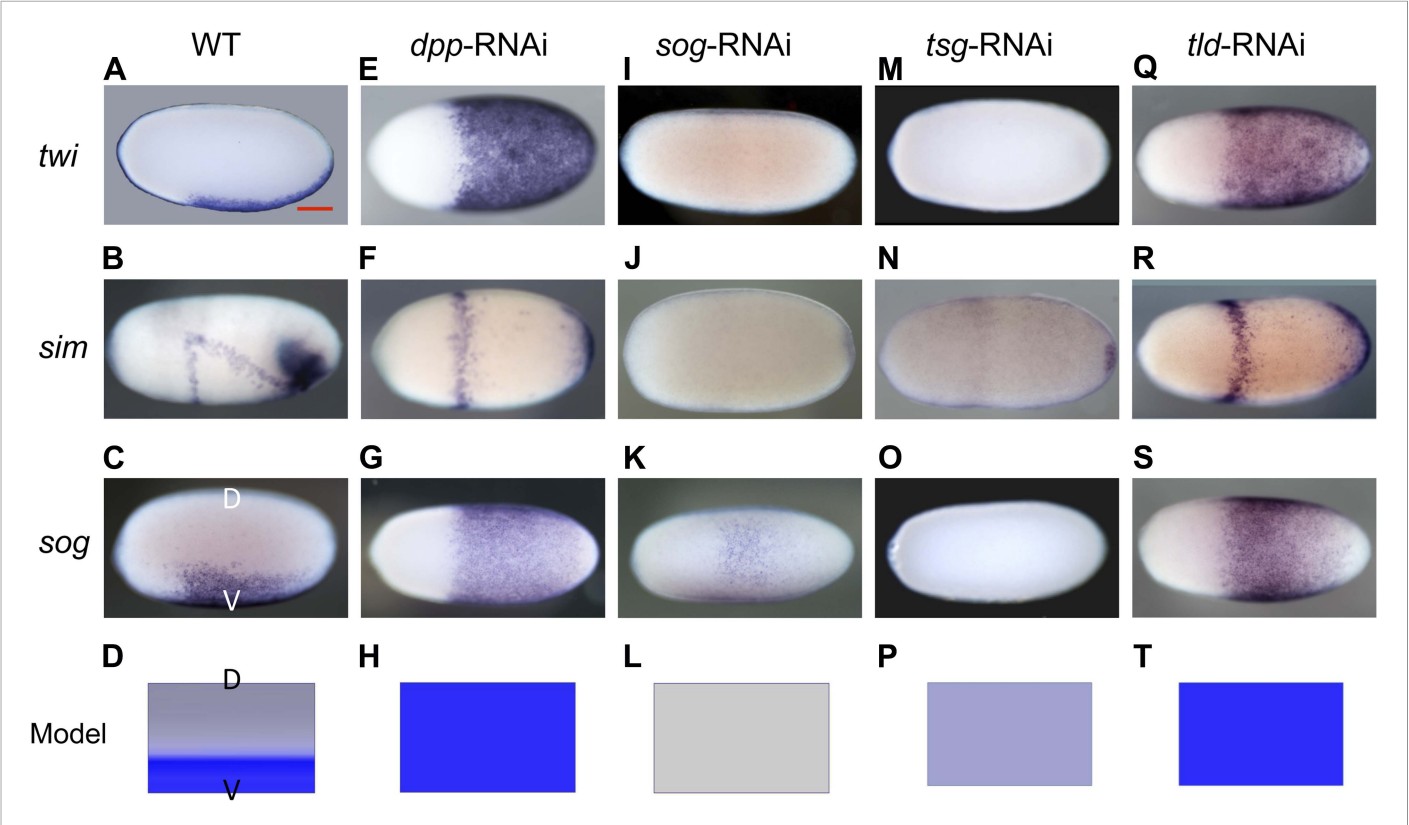

**Figure 2**. Knockdown (KD) of BMP signaling components results in completely ventralized (*dpp*-, *tld*-RNAi) or completely dorsalized (*sog*-, *tsg*-RNAi) embryos. Expression of *twi* (**A**, **E**, **I**, **M**, **Q**), *sim* (**B**, **F**, **J**, **N**, **R**) and *sog* (**C**, **G**, **K**, **O**, **S**) in wild type (wt) embryos (**A–C**), *dpp*-RNAi embryos (**E–G**), *sog*-RNAi embryos (**I–K**), *tsg*-RNAi embryos (**M–O**) and *tld*-RNAi embryos (**Q–S**) monitored by whole mount in situ hybridization (ISH). The view is lateral with the dorsal side pointing up (**A–C**), ventral (**K**), or not determined as the expression is DV-symmetric (**E–G**, **I**, **J**, **M–O**, **Q–S**). Embryos are at the blastoderm stage (~26–32 hpf: **A**, **C**, **E–G**, **I–K**, **M**, **O**, **Q**, **S**), or at the beginning of anatrepsis (posterior invagination of the embryo, ~33–37 hpf) (**B**). Scale bar (**A**) corresponds to 200 μm. For phenotype frequencies and confirmation of KD see *Figure 2—figure supplement 2* and *Figure 5—figure supplement 1*. (**D**, **H**, **L**, **P**, **T**) Simulations of the reaction diffusion system described in *Box 1* on a two-dimensional cylinder (*Figure 10*). Depicted is one half of the cylinder surface stretching from the dorsal (D) to the ventral (V) midline. Blue: *sog* expression ($\eta$). Gray: BMP concentration ($b$). (**D**) In wt *sog* expression is confined to a ventral stripe. (**H**) Loss of BMP ($b = 0$) leads to uniform derepression of *sog*. (**L**) Loss of *sog* ($s = 0$) leads to uniformly high levels of BMP. (**P**) Loss of Tsg was modeled by assuming that no Sog-BMP complexes are formed ($k_+ = 0$). This results in high BMP signaling throughout the embryo. (**T**) Loss of Tld was modeled by reducing the degradation constant of Sog ($\alpha_s$) by 90%. As Sog-BMP complexes are not degraded, BMP is not released, causing uniform derepression of *sog*.

The following figure supplements are available for figure 2:

**Figure supplement 1**. Expression of BMP signaling components during blastoderm.

**Figure supplement 2**. Phenotype frequencies after parental RNAi.

embryos pMAD accumulates more strongly in nuclei of dorsal than of ventral cells (*Figure 3A,F*; nuclear density can be used to distinguish dorsal and ventral regions of the embryos, *Figure 3—figure supplement 1*). Over time this asymmetry is enhanced. At the beginning of gastrulation, high levels of pMAD are restricted to the dorsal 30% of the egg circumference with sharp lateral borders (*Figure 3—figure supplement 1*). Within the domain of high nuclear concentrations the pMAD distribution is flat; i.e., it lacks the sharp peak along the dorsal midline which has been observed in *Drosophila* and *Nasonia* and is also less graded than the pMAD profile of *Tribolium* (*Dorfman and Shilo, 2001*; *van der Zee et al., 2006*; *Özüak et al., 2014b*).

Having established these four markers of distinct DV domains, we first analyzed the role of BMP signaling in *Oncopeltus*. Nuclear pMAD accumulation was largely abolished by knockdown (KD), via parental RNAi targeting the ortholog of *Drosophila* BMP ligand *dpp* (*O'Connor et al., 2006*). This

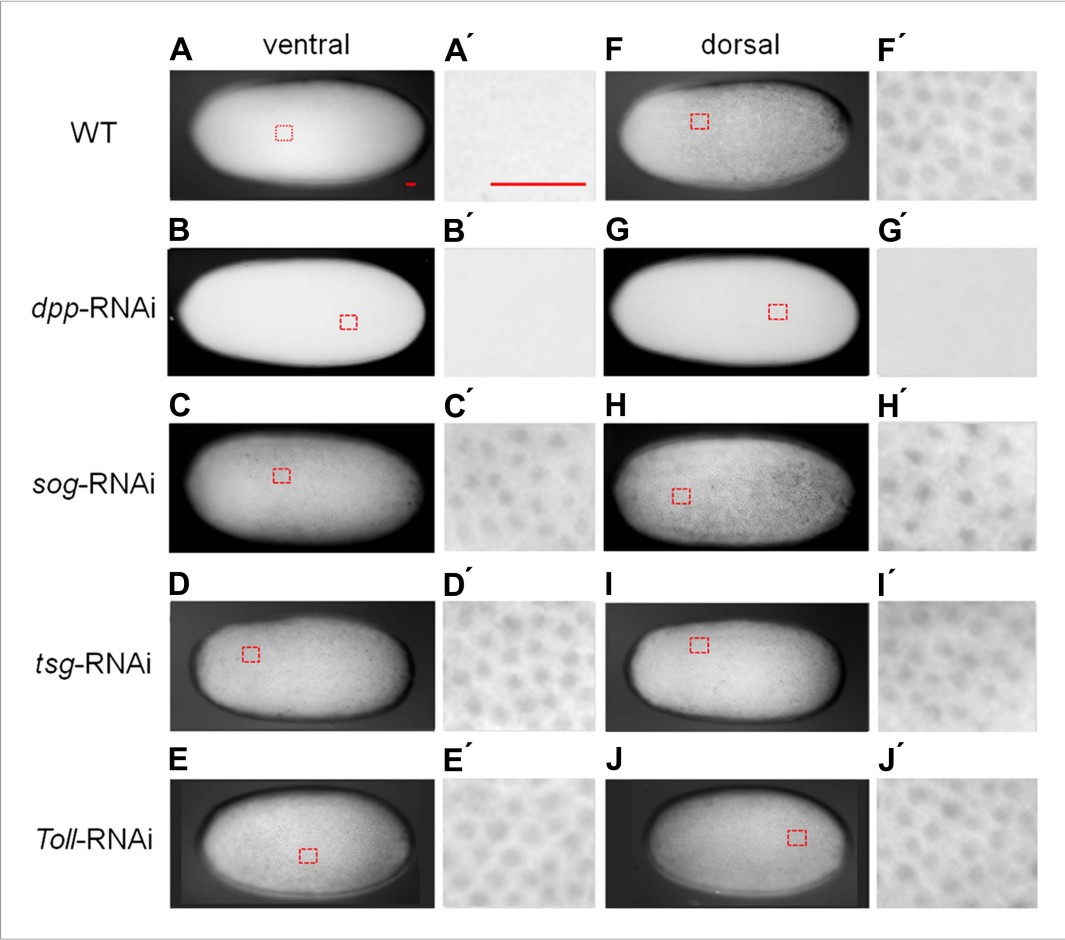

**Figure 3**. BMP signaling activity is uniformly abolished or expanded in ventralized or dorsalized phenotypes, respectively. pMAD distribution in blastoderm stage (26–32 hr post fertilization, hpf) wt (**A**, **A'**, **F**, **F'**), *dpp*-RNAi (**B**, **B'**, **G**, **G'**), *sog*-RNAi (**C**, **C'**, **H**, **H'**), *tsg*-RNAi (**D**, **D'**, **I**, **I'**) and *Toll1*-RNAi (**E**, **E'**, **J**, **J'**) embryos. For each embryo a ventral and a dorsal view, or views from opposite sides if DV polarity is lacking (**B–G**, **D-J**) are shown. Magnified surface views to the right of each embryo (x') reveal the presence or absence of pMAD in individual nuclei. The scale bar (**A**, **A'**) corresponds to 50 μm. For identifying the polarity of the DV axis and for BMP signaling activity during later development see *Figure 3—figure supplement 1*.

The following figure supplement is available for figure 3:

**Figure supplement 1**. Nuclear density and late pMAD distribution identify the dorsal side of *Oncopeltus* blastoderm embryos.

result confirms the specificity of our pMAD staining and demonstrates that the KD leads to a severe reduction of BMP signaling in the early embryo (*Figure 3B,G*).

Strikingly, this reduction of BMP signaling results in a massive expansion of *twi* and *sog* expression around the entire embryonic circumference (*Figure 2E,G*, for phenotype frequencies see *Figure 2—figure supplement 2*). The loss of lateral expression of *sim* (*Figure 2F*) shows that all fates dorsal to the mesoderm are lacking, indicating that the embryo is completely ventralized. Thus, BMP signaling is required in *Oncopeltus* to restrict the ventral-most, mesodermal cell fate. Absence of BMP signaling leads to a complete loss of DV polarity, which is not recovered during later stages of development (*Figure 4D–F*), a phenotype so far not known from other insects where BMP signaling either has no influence on the mesoderm (*van der Zee et al., 2006*; *Lynch and Roth, 2011*) or partially suppresses mesodermal cell fates (*Özüak et al., 2014b*).

Given the striking expansion of the ventral-most cell fate upon loss of BMP signaling in *Oncopeltus*, we wondered how ectopic BMP signaling would affect DV patterning. For this purpose we knocked

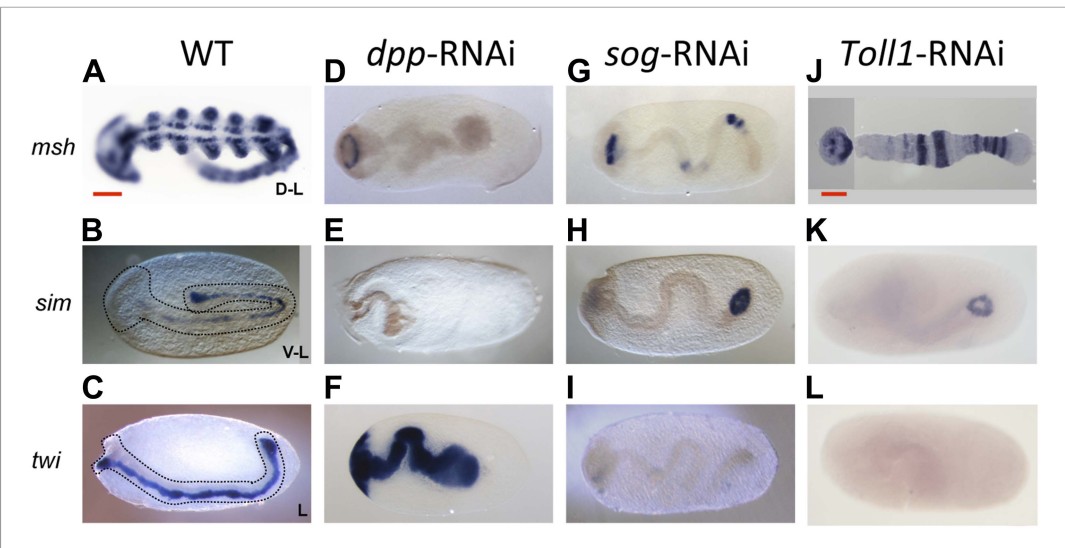

**Figure 4**. Late phenotypes of dpp, *sog* and *Toll1* KD embryos. Expression of *msh* (top row), *sim* (center row), and *twi* (bottom row) in wt (**A–C**), *dpp*-RNAi (**D–F**), *sog*-RNAi (**G–I**) and *Toll1*-RNAi (**J–L**) embryos monitored by ISH. The anterior of the embryo is on the left. Embryos are at the germ band stage (~40–48 hpf). **msh**: in wt germ band stage embryos *msh* is expressed in the dorsal-most part of the CNS and in the mesoderm of the limb buds (dorsal-lateral view). *dpp*-RNAi germ band embryos lack *msh* expression except for an anterior domain. *sog*- or *Toll1*-pRNAi embryos have a tube-like appearance lacking mesoderm and limb buds. Along these tubes *msh* is either not expressed or it is expressed at uniform levels around the entire circumference. This indicates that the ectoderm of *sog*- and *Toll1* KD embryos is dorsalized either at the level of the dorsal non-neurogenic or the dorsal-most neurogenic ectoderm. **sim**: in wt germ band stage embryos *sim* is expressed along the ventral midline (ventral-lateral view). Upon *dpp*-, *sog*- or *Toll1*-pRNAi, *sim* expression is lacking except for a ring of expression at the posterior tip of the growth zone in *sog*-RNAi and *Toll1*-RNAi embryos. This indicates that the ventral neuroectoderm is lost in these KD embryos. **twi**: in germ band stage embryos *twi* is expressed in the invaginated mesoderm, which forms initially a cord within the embryo (lateral view). In *dpp*-RNAi embryos *twi* is expressed in the entire germ band indicating complete mesodermalization. In *sog*-and *Toll1*-RNAi embryos *twi* is not expressed. This, in addition to the loss of *sim* expression, indicates that *sog* and *Toll1* KD embryos consistently lack ventral cell fates along their entire AP axis. Scale bar corresponds to 200 µm.

down *sog* and *twisted gastrulation* (*tsg*). Sog (Chordin) is known from *Drosophila* and several vertebrates to inhibit BMP signaling via ligand sequestering (***Little and Mullins, 2006***; ***O'Connor et al., 2006***). For Tsg both anti- and pro-BMP functions have been observed (***Wang and Ferguson, 2005***; ***Little and Mullins, 2006***; ***Nunes da Fonseca et al., 2010***; ***Özüak et al., 2014b***). In *Oncopeltus*, the KD of these two genes causes elevated levels of pMAD at the ventral side and frequently leads to a uniform distribution of pMAD around the embryonic circumference, indicating that the asymmetry of BMP signaling depends on *sog* and *tsg* (***Figure 3C,D,H,I***). The levels of pMAD around the entire circumference are similar to the levels found at the dorsal side of wild type (wt) embryos. The KD embryos show a complete loss of the mesoderm and mesectoderm, as demonstrated by the loss or strong reduction of *twi*, *sog* and *sim* expression (***Figure 2I–K,M–O***; for phenotype frequencies see ***Figure 2—figure supplement 2***). This further indicates dorsalization and a lack of DV polarity when BMP signaling is uninhibited.

During later development, DV polarity is not recovered: gastrulation and all subsequent morphogenetic movements lack DV asymmetry (***Figure 4G–I***). Thus, in *Oncopeltus*, in contrast to all other insects analyzed so far (***Lynch and Roth, 2011***), BMP signaling has to be suppressed ventrally by Sog (in conjunction with Tsg) to allow polarization of the DV axis and specification of ventral cells. The essential role of Sog is supported by the consequences of a depletion of Tolloid (Tld), which is known to cleave and inactivate Sog and thereby to release bound BMP ligands (***O'Connor et al., 2006***). As with the *dpp* KD, *tld* KD leads to a complete ventralization of the embryo, indicating that BMP ligands are largely (or completely) sequestered in inhibitory Sog-BMP complexes in the absence of Tld (***Figure 2Q–S***).

Besides the complete loss of embryonic DV polarity, the KD phenotypes reveal an interesting regulatory feature of the BMP network in *Oncopeltus*. *sog* expression is expanded or suppressed by reducing (*dpp* KD) or expanding (*tsg* KD) BMP activity, respectively, demonstrating that BMP signaling negatively regulates its own antagonist in *Oncopeltus*. This was never observed in holometabolous insects, where *sog* expression either exclusively depends on Toll signaling lacking feedback control by BMP (*Drosophila* and *Tribolium*) (*Jazwinska et al., 1999*; *van der Zee et al., 2006*) or is absent (*Nasonia*) (*Özüak et al., 2014b*). However, in spiders (*Akiyama-Oda and Oda, 2006*), vertebrates (*De Robertis and Kuroda, 2004*), and sea anemones (*Saina et al., 2009*) the *sog* homolog *chordin* is directly or indirectly repressed by BMP signaling, indicating that the BMP network of *Oncopeltus* exhibits a regulatory property that is ancestral for animals.

As all cell fates along the DV axis are affected by BMP signaling, we wondered whether Toll signaling is even required for DV patterning in *Oncopeltus*. A recent study in another hemipteran, *Rhodnius prolixus* has provided evidence that Toll signaling plays a role in DV patterning (*Berni et al., 2014*). In *Oncopeltus*, KD of the *Toll1* ortholog resulted in loss of *twi* and *sim* expression (*Figure 5E,F*) and *sog* expression was completely lacking in 38% of the mid and late blastoderm stage embryos (*Figure 5G*; for phenotype frequencies and confirmation of KD see *Figure 5—figure supplement 1*). As expected, this leads to high uniform levels of pMAD around the entire embryo circumference (*Figure 3E,J*). Expression analysis of germ band stage embryos confirms that the *Toll1* KD embryos are dorsalized and lack all DV polarity (*Figure 4J–L*). KD of other downstream components of Toll signaling (Myd88, Pelle, Tube-like kinase) leads to identical phenotypes (data not shown). Two homologs of NF-κB/Dorsal (*Of-dl1* and *Of-dl2*), the transcription factor acting downstream of Toll signaling (*Stein and Stevens, 2014*), were identified. KD of both caused a loss of ventral gene expression, albeit to varying degrees, indicating at least partially redundant functions (shown for *dl1*: *Figure 5H–J*; *Figure 5—figure supplement 1*). Taken together, interfering with Toll signaling leads to dorsalized phenotypes, which closely resemble those produced by KD of *sog* and *tsg*.

However, loss of Toll signaling also has consequences not observed by manipulating the BMP pathway. This becomes apparent by looking at marker genes expressed in head anlagen like *muscle-specific homeobox* (*msh*). In wt blastoderm embryos *msh* is expressed in a stripe with a sharp anterior border and a posterior border positioned at approximately 60% egg length (0% is the posterior pole). After *Toll1* and *dl1* KD the posterior *msh* border is shifted anteriorly, typically to 80% egg length, and the stripe expands towards the anterior tip of the embryo (*Figure 6B,C*). This does not occur in dorsalized embryos after *tsg* KD (*Figure 6D*). Using other markers, AP shifts have also not been seen in ventralized embryos after *dpp* and *tld* KD (see anterior *sim* stripe in *Figure 2F,R*). We therefore assume that Toll, unlike BMP signaling, is not only dedicated to DV patterning in *Oncopeltus*, but also contributes to specifying the AP axis. A role for Toll in positioning the embryo along the AP axis has recently been suggested for the hemipteran *Rhodnius* (*Berni et al., 2014*).

Since *twi* and *sog* are completely dependent on Toll signaling for their activation in all studied holometabolous insects, we hypothesized that the expansion of *twi* and *sog* in *dpp* KD was due to a corresponding expansion of Toll signaling in the absence of BMP-dependent repression. To test this hypothesis, we produced embryos simultaneously lacking Toll and BMP signaling. To our surprise the *Toll1 dpp* double KD embryos showed uniform *twi* and *sog* expression along the embryonic circumference (*Figure 5L,N*; *Figure 5—figure supplement 1*), the same as the *dpp* single KD (*Figure 2E,G*). However, in contrast to the single KD of *dpp*, the double KD embryos also show an expansion and/or shift of the *sog*, *twi* and (anterior) *sim* domains towards the anterior pole (compare *Figure 2E–G* and *Figure 5L–N*). This is likely due to the additional role of Toll signaling in anterior patterning and allows for an unambiguous distinction between double and single KD embryos (additional confirmation by RT-PCR, *Figure 5—figure supplement 1*). Our results suggest that DV patterning genes that require Toll signaling for expression in other insects can be activated in the absence of Toll signaling in *Oncopeltus*. As these genes are repressed by elevated BMP activity their state of expression seems to be mainly controlled by different levels of BMP signaling. This leads to the crucial question: What then is Toll's role within the DV patterning system of *Oncopeltus* if Toll is neither strictly required to activate ventral genes nor to prevent their repression?

To address this question we carefully studied the dynamics of *sog* expression in wt and *Toll1* KD embryos. Interestingly, *sog* transcription is activated ubiquitously in early blastoderm embryos (*Figure 5R*). Only later is *sog* expression enhanced at the ventral side, while weak *sog* expression is still seen dorsally (*Figure 5T,V*). Finally, during mid-blastoderm (25–28 hpf) the typical *sog* expression

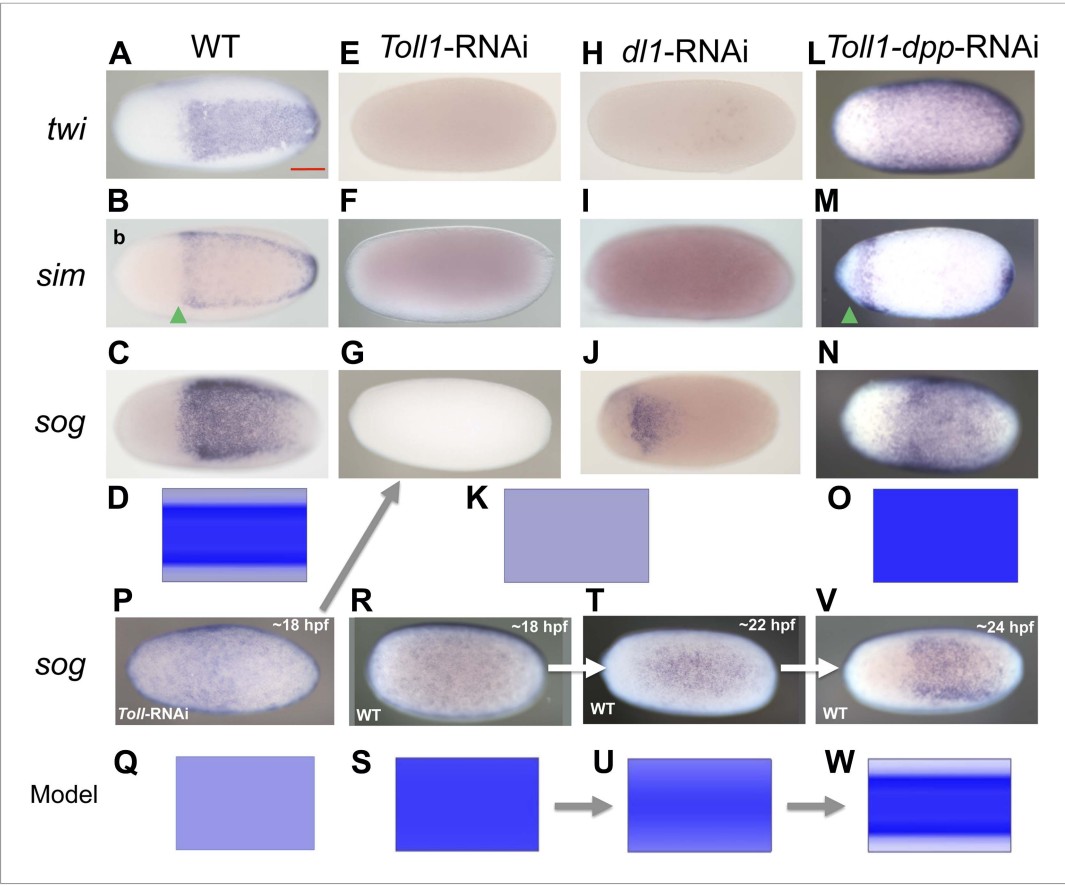

**Figure 5**. BMP signaling is epistatic to Toll signaling in *Oncopeltus*. Expression of *twi* (**A**, **E**, **H**, **L**), *sim* (**B**, **F**, **I**, **M**), *sog* (**C**, **G**, **J**, **N**, **P**, **R**, **T**, **V**) in wt embryos (**A**–**C**, **R**, **T**, **V**), *Toll1*-RNAi embryos (**E**–**G**, **P**), *dl1*-RNAi embryos (**H**–**J**) and *Toll1*-*dpp*-RNAi embryos (**L**–**N**) monitored by ISH. The view is ventral (**A**–**C**, **J**, **T**, **V**), or not determined as the expression is DV symmetric (**E**–**G**, **H**, **I**, **L**–**N**, **P**, **R**). Embryos are at the blastoderm stage (**A**–**C**, **E**–**G**, **H**–**J**, **L**–**N**: 26–32 hpf; **P**–**V** see figure labels). Green arrowheads mark the anterior border of *sim* expression. The scale bar (**A**) corresponds to 200 µm. For phenotype frequencies and confirmation of KD see *Figure 5—figure supplement 1*. (**D**, **K**, **O**, **Q**, **S**, **U**, **W**) Simulations of the reaction diffusion system described in *Box 1* on a two-dimensional cylinder (*Figure 10*). Depicted is the ventral part of the cylinder. Blue: *sog* expression level ($\eta$). Gray: BMP concentration (*b*). (**D**) wt: *sog* expression is confined to a ventral stripe. (**K**) Upon loss of active NF-κB/Dorsal (*d* = 0) due to either KD of *Toll1* or KD of *dl1*, early activation of *sog* (**P**) is insufficient to initiate patterning resulting in uniformly high BMP signaling. (**O**) Upon simultaneous loss of Dorsal (*d* = 0) and BMP (*b* = 0) *sog* activation is possible despite lack of NF-κB/Dorsal; however, activation is uniform. (**Q**) *sog* activation at early stages in the absence of Toll signaling (*d* = 0). This reflects $\eta_o$, NF-κB/Dorsal-independent *sog* activation (*Box 1*). (**S**, **U**, **W**) Developmental progression of *sog* activation ($\eta$) during blastoderm stages.

The following figure supplement is available for figure 5:

**Figure supplement 1**. Phenotype frequencies and transcript levels after RNAi.

domain is established, with high levels in the ventral 40% and no detectable expression in the dorsal 60% of the germ rudiment (*Figures 5C, 2C*). Early *Toll1* KD embryos show uniform expression of *sog* (*Figure 5P*), which disappears during later stages (*Figure 5G*). Thus, Toll is not required to initiate *sog* expression, but rather to enhance its expression ventrally. A weakly asymmetric Toll gradient might suffice to fulfill this function.

To get a first impression of the shape of the Toll signaling gradient in *Oncopeltus*, we analyzed the expression of *cactus* (*cact*) genes encoding the insect I-κB homologs which bind to NF-κB/Dorsal and prevent nuclear transport (*Bergmann et al., 1996*). The transcriptional activation of *I-κB* genes by Toll

**Figure 6**. Toll signaling affects AP patterning. Expression of *msh* is monitored by ISH in blastoderm embryos. The view is lateral (**A**), or not determined as the expression is DV symmetric (**B**–**D**). The red arrowheads mark the posterior border of *msh* expression which is positioned at approximately 60% egg length (0% posterior pole) in wt (**A**) and *tsg* KD (**D**) embryos. In *Toll1* and *dl1* KD embryos, the *msh* domain expands to the anterior tip of the embryo and its posterior border is shifted anteriorly (to approximately 80% egg length).

signaling appears to be an ancestral negative feedback loop essential for attenuating the innate immune response triggered by Toll (*Hoffmann et al., 2002*). During *Drosophila* and *Tribolium* DV patterning *cact* is an early target gene of Toll signaling expressed in regions of high nuclear NF-κB/ Dorsal concentrations (*Sandmann et al., 2007*; *Nunes da Fonseca et al., 2008*). The same appears to apply to *Nasonia* where *cact* expression is restricted to a narrow stripe straddling the ventral midline indicating a highly refined pattern of Toll activity (*Buchta et al., 2013*; *Özüak et al., 2014b*). The *Oncopeltus* genome harbors six *cact* paralogs, four of which are expressed during blastoderm stages (*Vargas Jentzsch et al., 2015*) (and data not shown). While *cact1, 2* and *4* show only weakly asymmetric expression (*Figure 7A–C*, and data not shown), *cact3* is expressed in a broad ventral domain encompassing 60–80% of the embryonic circumference with graded borders toward the dorsal side (*Figure 7E–G,I,J*). The expression of *cact3* does not refine into a more narrow domain, but remains broad during later blastoderm stages. *Toll1* KD embryos lack (or show reduced) *cact1* and *cact3* expression, confirming the regulatory link known from other insects (*Figure 7D,H*). These observations support the notion that in *Oncopeltus*, Toll signaling is transiently active almost along the entire DV axis and forms a shallow gradient with lower levels in the dorsal half.

In sum, our empirical observations lead to the following model for DV patterning in *Oncopeltus* (*Figure 1*, *Box 1*). We posit that during early blastoderm stages, weak uniform BMP signaling is balanced by the uniform Toll-independent production of the BMP inhibitor Sog. A shallow Toll signaling gradient breaks this symmetry by enhancing *sog* expression at the ventral side. This leads both to ventral suppression of BMP signaling, and to a flux of BMP-Sog complexes to the dorsal side. Subsequently, the Tld-dependent cleavage of Sog releases BMP and hence increases BMP signaling. Since BMP signaling represses *sog* expression, the asymmetry initiated by Toll is dynamically enhanced.

To investigate the dynamics of the Sog/BMP system we constructed a minimal reaction-diffusion model as in previous work in *Drosophila* (*Eldar et al., 2002*) (*Box 1*, 'Materials and methods'). In this model the rate of *sog* expression combines NF-κB/Dorsal-independent and NF-κB/Dorsal-dependent activation with repression of *sog* by BMP. Parameter settings were selected such that NF-κB/Dorsal-dependent *sog* activation is necessary in order to initiate patterning (*Figures 8, 9* and *Table 1*). This mode of *sog* activation tightly links DV axis formation to egg polarity via Toll signaling (*Stein and Stevens, 2014*) and provides stability against random fluctuations (*Box 1*, 'Materials and methods', *Figure 9*).

Two-dimensional simulations on a cylinder representing the trunk region of the ellipsoid embryo show that the model robustly replicates the formation of stripe-like *sog* expression domains (*Figure 2D*; *Figure 5D,W*; *Figure 10*). Moreover, the model correctly recovers the steady state of *sog* expression and BMP distribution in KD embryos (*Figure 2H,L,P,T*; *Figure 5K,O*), including the dynamics of *sog* expression in *Toll1* KD (*Figure 5Q,K*), as well as wt embryos (*Figure 5S,U,W*).

Simulations also reveal that even weakly polarized NF-κB/Dorsal gradients result in sharp BMP signaling profiles (*Box 1*, *Figure 9*). The final patterning output is robust with regard to variation in width of the NF-κB/Dorsal gradient along the AP axis (*Figure 11*). Likewise, raising the NF-κB/Dorsal concentration above the critical threshold for *sog* activation along the entire DV axis had no impact on the patterning output (*Figure 9*).

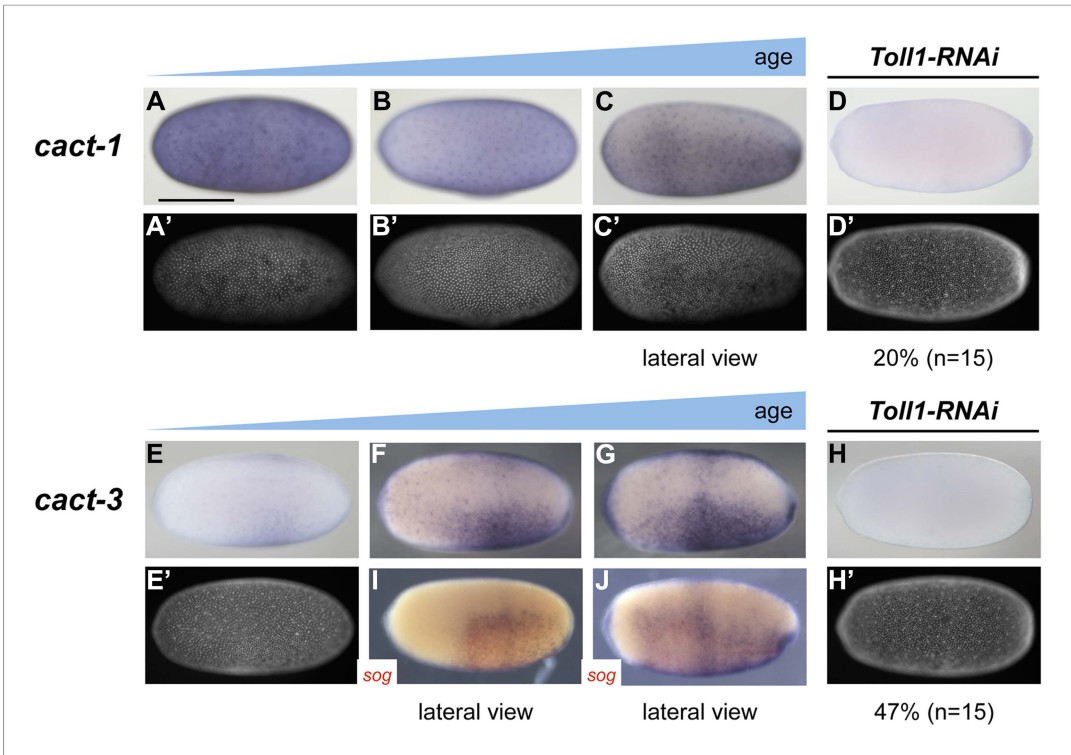

**Figure 7**. Expression of *cact1* and *cact3*. Expression of *cact1* (**A–D**) and *cact3* (**E–H**) are monitored by ISH with embryos at early to late blastoderm stages (20–32 hpf). (**A'–D'**, **E'**, **H'**) SYTOX Green staining shows nuclear density to determine developmental stage. (**A**, **B**) *cact1* expression is initiated evenly. (**C**) With proceeding development *cact1* expression vanishes from the dorsal side. (**D**) 20% of *Toll1* KD embryos lack *cact1* expression. The remainder show reduced expression compared to wt. (**E**) *cact3* expression is initiated uniformly along the DV axis between 20% and 60% egg length. (**F**, **G**) In older blastoderm stages *cact3* is expressed in a broad domain encompassing 60–80% of the egg circumference. (**H**) 47% of *Toll1* KD embryos lack *cact3* expression. The remainder show reduced expression compared to wt. (**I**, **J**) Double ISH for *cact3* (blue) and *sog* (red) confirms that *cact3* is expressed ventrally and that its domain expands more dorsally than the *sog* domain.

Such weakly polarized, broad NF-κB/Dorsal distributions in conjunction with the proposed dynamic BMP signaling system can explain embryonic twinning induced by egg fragmentation in another hemipteran species, the leaf hopper *Euscelis plebejus* (*Sander, 1971*). In the course of his experiments, Sander produced dorsal and ventral egg fragments during early development (preblastoderm) using a guillotine which completely separated the two egg halves. Complete germ band embryos developed within each half (*Box 1*). The model proposed here can reproduce such regulative behavior. A split along the DV axis prior to the initiation of patterning prevents diffusion from the ventral to the dorsal half. Thus, BMP acting as a long-range inhibitor of *sog* expression cannot travel from the ventral to the dorsal half to suppress *sog*. Since the NF-κB/Dorsal gradient extends to the dorsal half, *sog* can be activated dorsally and initiates a second round of patterning. Consequently, *sog* domains and BMP gradients are produced independently in each half despite the NF-κB/Dorsal gradient itself not having been altered (*Box 1*). The sizes of the *sog* and BMP domains are adjusted to the dimensions of the egg halves implying that the patterning process shows almost perfect scaling. Furthermore, the predicted orientation of the embryos with ventral sides pointing to the dorsal egg pole (after axis inversion through anatrepsis, [*Panfilio, 2008*]) corresponds to the most frequently observed experimental outcome (*Sander, 1971*). Thus, our model represents a minimal BMP/Sog (Chordin) system that exhibits self-organized DV patterning and explains a striking result from classical insect embryology. The only requirement for the NF-κB/Dorsal gradient is that it extends into the dorsal half of the embryo. The expression of *cact* suggests that this condition is fulfilled in *Oncopeltus*. Unfortunately, the mechanical properties of *Oncopeltus* eggs prevents egg fragmentation to directly investigate the potential for twinning.

# Box 1.

We built a reaction-diffusion model of the BMP/Sog system based on (i) inhibition of *sog* expression by BMP, (ii) *sog* transcriptional activation by NF-κB/Dorsal, (iii) the binding of Sog to BMP and (iv) the rapid diffusion of the Sog-BMP complex.

A simple Michaelis–Menten model of *sog* regulation gives the rate of *sog* expression

$$\eta_s(b, d) = \frac{\eta_0 + \eta_1\, d/d_0}{(1 + b/b_0)(1 + d/d_0)},$$

in terms of the local concentrations b (BMP) and d (NF-κB/Dorsal). $\eta_0$ is the rate of *sog* expression in the absence of NF-κB/Dorsal and BMP, $\eta_1 > \eta_0$ is the asymptotic rate of *sog* expression at high concentrations of NF-κB/Dorsal and in the absence of BMP. This model can exhibit an instability of the homogeneous state (***Box figure 1***). Consider a small 'seed' of elevated Sog concentration arising from the polarity cue provided by NF-κB/Dorsal. Sog molecules bind BMP and the complexes diffuse away quickly, leading to a depletion of BMP. Since BMP represses *sog*, this leads to a local increase of *sog* expression, causing the seed to grow. In the steady state, there is a region of high Sog levels (where BMP diffuses away quickly due to complex formation) around the original seed, and a region of high BMP levels away from the seed, where *sog* is repressed by BMP.

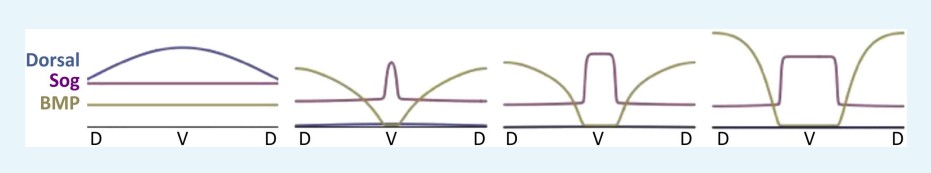

**Box figure 1**. Temporal progression of pattern formation. Simulated concentration profiles of NF-κB/Dorsal, Sog and BMP are plotted at successive time points, starting from a broad peak of NF-κB/Dorsal (left to right: t = 0 hr, 4 hr, 6 hr and steady state; see ***Table 1*** for parameter values). The x-axis shows the circumference of the embryo, with the dorsal and ventral sides marked as D and V, respectively.

The instability turns out to be controlled by the level of NF-κB/Dorsal. A threshold amount of NF-κB/Dorsal is required initially to build a stripe of high Sog concentration near the initial NF-κB/Dorsal maximum (***Figures 8, 9***). This effect also leads to the phenomenon of twinning: if the amount of NF-κB/Dorsal is above the threshold in both halves of the embryo, a cut along the DV axis can lead to the formation of a stripe in each half (***Box figure 2***).

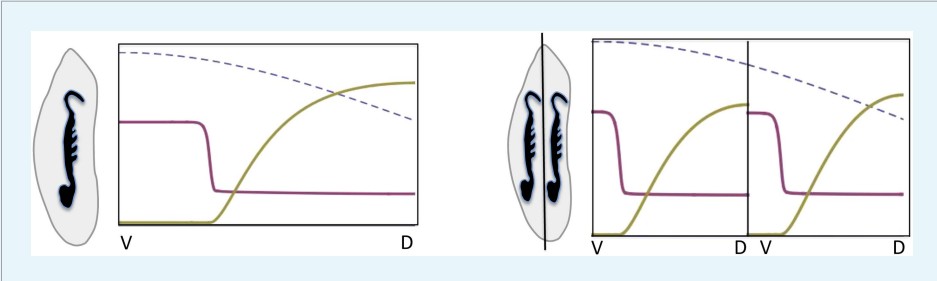

**Box figure 2**. Embryonic twinning in *Euscelis*. Simulation of wt (left) showing Sog (red) and BMP (green) protein concentration profiles along the DV axis at steady state given initial NF-κB/Dorsal protein levels (blue: dashed line). A schematic drawing on the left shows a wt *Euscelis* embryo at the germ band stage. Simulations after the DV axis is split in two halves (right) result in the formation of one BMP gradient in each half. Thus, the proposed model can account for the embryonic twinning observed in *Euscelis* after production of dorsal and ventral egg fragments shown schematically on the left (***Sander, 1971***).

The dynamics of our model are similar to those of activator-inhibitor models of pattern formation (*Turing, 1952*; *Gierer and Meinhardt, 1972*). However, in our model there is no explicit activator. Instead, the patterning mechanism emerges from the transport of the Sog-BMP complex away from areas with elevated Sog concentration, leading to a derepression of *sog* by removal of BMP.

## Discussion

This paper describes three key properties of DV patterning in *Oncopeltus*. (1) BMP signaling represses the transcription of its extracellular inhibitor *sog*, a feature so far not observed in other insects, but characteristic for spiders and vertebrates. (2) BMP signaling patterns the entire DV axis by acting repressively on ventral cell fates. This is similar to other bilaterian animals but within insects has so far only been seen in the hymenopteran *Nasonia*, a representative of the most basally branching holometabolous insect order. (3) Toll is neither a long-range morphogen, as in *Drosophila* and *Tribolium*, nor a local inducer of particular cell fates (mesoderm and mesectoderm), as in *Nasonia*. In *Oncopeltus*, Toll apparently only acts via its control of BMP signaling. In the following, we discuss these three points and their implications for the evolution of DV patterning in insects.

### BMP represses *sog* in *Oncopeltus*

In *Drosophila* and *Tribolium*, the expression of *sog* is dependent on Toll signaling (*van der Zee et al., 2006*; *Liberman and Stathopoulos, 2009*; *Nunes da Fonseca et al., 2010*). *sog* can be neither activated in absence of Toll (*Stathopoulos and Levine, 2004*) nor can it be repressed by ectopic BMP signaling (*Jazwinska et al., 1999*). However, a binding site for transcription factors acting downstream of BMP signaling (Schnurri-Mad-Medea sites) has recently been identified within the proximal enhancer of *Drosophila sog* (*Ozdemir et al., 2014*). Its functional significance is not known, but it might represent an evolutionary relict of the regulatory logic we have observed in *Oncopeltus* in which inhibitory BMP signaling is essential for defining the *sog* expression domain. A negative feedback of BMP on sog/chordin expression is familiar from many animal phyla. It has even been found in sea anemones, predating the emergence of the bilaterian body plan (*Saina et al., 2009*;

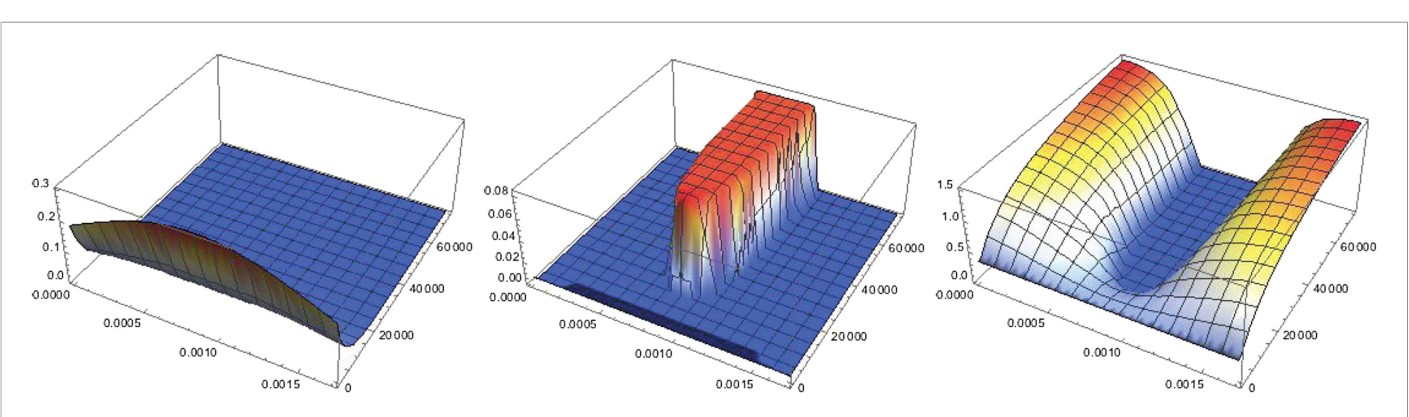

**Figure 8**. Dynamics of pattern formation. Each plot shows the concentration of a particular protein species in space (*x* running from 0 to $l_x$ along the front of the plot parameterizing the circumference of the cylinder) and time (running towards the back). Left: the concentration of NF-κB/Dorsal shows a broad Gaussian profile that decays to zero with time. Center: starting from a uniform distribution a region of high Sog concentrations forms where the initial distribution of NF-κB/Dorsal had its maximum. Right: BMP is depleted where Sog levels are high. Initial conditions are $b(x, t = 0) = 0.32$, $s(x, t = 0) = 0.01$, $c(x, t = 0) = 0.14$, $d(x, t = 0) = D_o \exp\left\{-\frac{1}{2}(2/l_x)^2(x - 2/l_x)^2\right\}$. Throughout the text $D_o = 0.3$, except in the twinning figure (**Box 1**), where $D_o = 1$ was used to ensure a sufficient amount of NF-κB/Dorsal in both halves of the embryo.

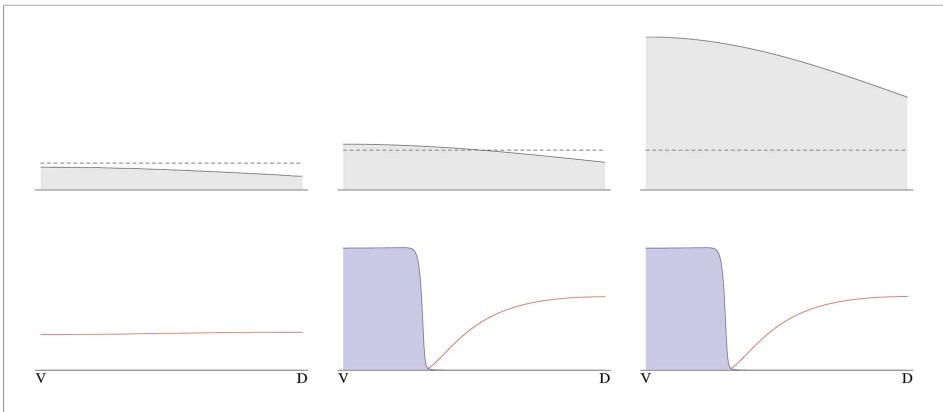

**Figure 9**. Pattern formation from different initial levels of NF-κB/Dorsal. The initial concentration gradient of NF-κB/Dorsal is shown on top (gray). Initial amplitudes of NF-κB/Dorsal are $D_o = 0.15$, $0.3$, $1$ from left to right, the dashed line indicates the threshold NF-κB/Dorsal concentration required for patterning. Below, steady-state levels of free BMP (red) and free Sog are shown (blue, rescaled to facilitate plotting on the same plot as BMP).

*Genikhovich et al., 2015*). For both hemichordates and basal chordates, evidence was provided that BMP suppresses *chordin*, directly or indirectly (*Lowe et al., 2006*; *Yu et al., 2007*). This applies also to spiders, which however have evolved a special strategy of DV axis formation that is radically different from most other animal phyla. In spiders the migration of the BMP expressing cumulus cells towards a symmetric ring of *chordin* expression breaks DV symmetry (*Akiyama-Oda and Oda, 2006*). Thus, spiders polarize the DV axis not by localizing inhibitor expression, but rather by localizing BMP.

Among the well-studied DV patterning systems, *Oncopeltus* can be best compared to vertebrates. In the zebrafish and the frog the negative regulation of *chordin* by BMP signaling is an important feature to explain normal patterning and axis duplication (twinning) after transplantations (*Oelgeschläger et al., 2003*; *De Robertis, 2009*; *Langdon and Mullins, 2011*; *Xue et al., 2014*). However, the networks involved in size regulation of the embryonic axis are more complex and require many additional components such as ADMP (antidorsalizing morphogenetic protein), a BMP-type ligand co-expressed with Chordin, and Sizzled, an antagonist of Tld that is co-expressed with BMP. ADMP and Sizzled have been implicated in scaling both experimentally and by modeling approaches (*Reversade and De Robertis, 2005*; *Ben-Zvi et al., 2008*; *Inomata et al., 2013*). No homologs of these genes were found in the *Oncopeltus* genome and transcriptomes (*Ewen-Campen et al., 2011*) (*Vargas Jentzsch et al., 2015*), and appropriate scaling occurred in our theoretical simulations of twinning, without the need to invoke such additional modulators.

Thus, the *Oncopeltus* system is surprisingly simple and may represent a minimal network able to support self-organized patterning. Our theoretical model emerged from modifications of equations that have been used to describe the formation of peak levels of BMP signaling along the dorsal midline in *Drosophila* (*Eldar et al., 2002*). The BMP signaling peak in *Drosophila* forms within a domain of uniform *dpp* and *tld* expression and depends on both the diffusion of

**Table 1**. Model parameters.

| | |
|---|---|
| $\eta$ | $2.8 \times 10^{-4}$ |
| $\eta_1$ | $2 \times 10^{-3}$ |
| $b_0$ | $0.2$ |
| $d_0$ | $1$ |
| $\eta_b$ | $4 \times 10^{-5}$ |
| $\alpha_s$ | $2 \times 10^{-3}$ |
| $\alpha_b$ | $5 \times 10^{-5}$ |
| $\alpha_c$ | $2 \times 10^{-4}$ |
| $K_+$ | $5$ |
| $k_-$ | $5 \times 10^{-5}$ |
| $D_s$ | $1.5 \times 10^{-13}$ |
| $D_b$ | $7.8 \times 10^{-13}$ |
| $D_c$ | $2.5 \times 10^{-9}$ |
| $l_x$ | $0.0017$ |

Units are arbitrary but are suggested to be seconds for time and meters for length.

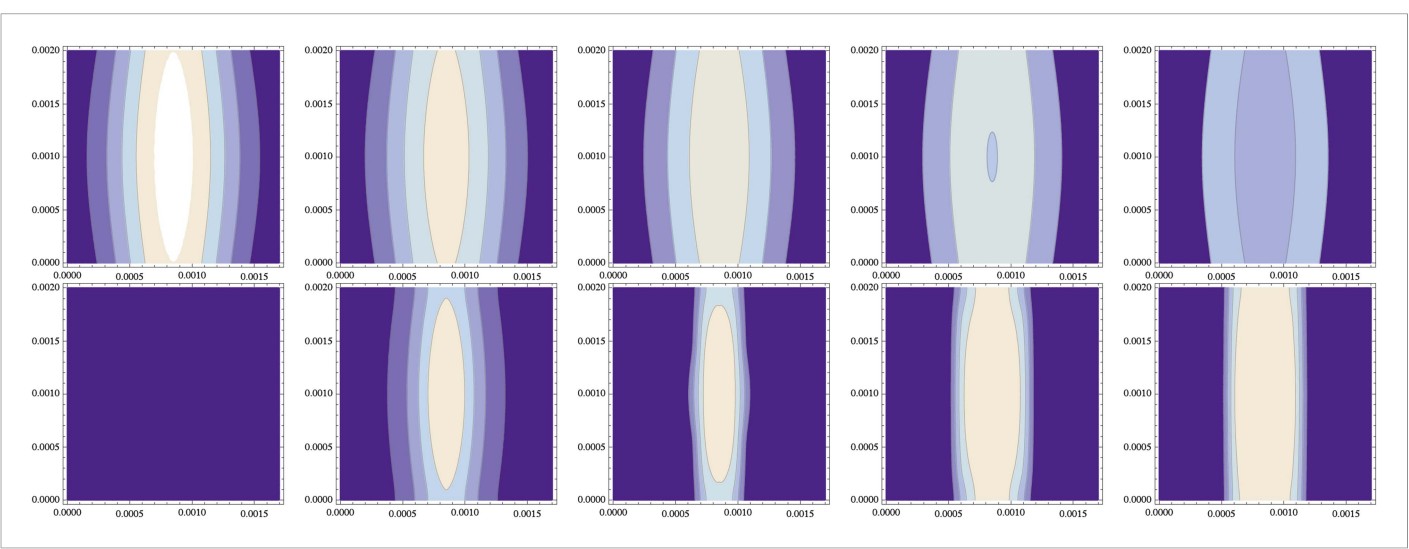

**Figure 10**. Pattern formation in two dimensions. Starting from a distribution of NF-κB/Dorsal with a broad maximum running in parallel to the cylinder's axis (bottom, shown in green), a stripe of high Sog concentration develops (top, Sog shown in blue, BMP shown in red). The figures show concentrations at times 0, 1000, 2000, 3000, 4000 from left to right.

Sog-BMP complexes towards the dorsal side (from the ventral source region of *sog*) and on the degradation of Sog by Tld to release active BMP ligands dorsally. The *Drosophila* system is static, that is, the regions of ventral *sog* expression and the abutting regions of *dpp* and *tld* are fixed by the NF-κB/Dorsal gradient. The early dynamics of the system are restricted to reaction diffusion processes. In *Oncopeltus*, on the other hand, the transcriptional feedback of BMP signaling on *sog* expression creates a situation in which the size of the *sog* expression domain itself is an outcome of the system dynamics and becomes largely independent from NF-κB/Dorsal.

In *Drosophila* and *Tribolium* the transport of BMP by Sog leads to dorsal BMP signaling levels that are higher than the signaling levels in the absence of Sog (***Dorfman and Shilo, 2001***; ***van der Zee et al., 2006***). This seems not to be the case in *Oncopeltus*, as the uniformly high BMP levels seen in *sog* or *tsg* KD embryos match the levels found on the dorsal side of control embryos (***Figure 3***). Furthermore, the BMP profile during normal development is flat, with a narrow and steep transition between uniform BMP signaling dorsally and the absence of BMP signaling ventrally. In our simulations the *sog* and BMP profiles are also flat, indicating that the negative feedback of BMP

**Figure 11**. Independence from the stripe of initial conditions (same data as *Figure 10*). (top) The initial distribution of NF-κB/Dorsal from *Figure 10* varies along the cylinder's axis (y-direction of this contour plot, with the x-direction describing the circumference) in both standard deviation and amplitude by about 10%; $d(x, t=0) = \exp\left\{-\frac{5^2}{2(1+0.1\sin(\pi y/l_y))^2}(x - 2/l_x)^2\right\}(1 + 0.1\sin(\pi y/l_y))$, and decays over time (time points 0, 1000, 2000, 3000, 4000 shown from left to right). (bottom) The resulting distribution of Sog (and correspondingly of BMP) becomes uniform along the cylinder axis.

signaling on *sog* expression might prohibit the formation of sharp peak levels of BMP signaling (*Box 1*, *Figures 8, 9*). Similarly, in vertebrates Chordin-mediated BMP transport does not markedly enhance BMP signaling levels, as *chordin* KD does not lead to lateralization, but rather to ventralization of the embryo (*Schulte-Merker et al., 1997*; *Oelgeschläger et al., 2003*).

An unusual feature of the *Oncopeltus* system is the strong anti-BMP function of Tsg. In *Drosophila* Tsg has a mild pro-BMP function that is independent of Sog (*Wang and Ferguson, 2005*). In *Tribolium*, Tsg is essential for all BMP activity in a Sog-independent manner (*Nunes da Fonseca et al., 2010*). The same holds true for *Nasonia,* which lacks Sog (*Özüak et al., 2014b*), suggesting that a Sog-independent pro-BMP function of Tsg might be ancestral for insects. To our surprise we observed the opposite in *Oncopeltus*, where the strong anti-BMP function of *tsg* is responsible for the similarity of the *sog* and *tsg* KD phenotypes (*Figures 2, 3*). Thus, Sog can exert its inhibitory effect on BMP only in the presence of Tsg. It will be interesting to find out whether this strong anti-BMP function of Tsg has a particular significance in a system where *sog* is repressed by BMP signaling.

We assume that patterning in *Oncopeltus* is initiated when NF-κB/Dorsal-dependent enhancement of *sog* expression at the ventral side exceeds a certain threshold. This conclusion is based on the analysis of embryos with incomplete KD of *sog* (*Figures 2K, 5J*). Such embryos frequently show an asymmetric pMAD distribution although they lack ventral gene (*twi*, *sim*) expression (*Figure 2—figure supplement 2*). Thus, ventral BMP signaling has to decrease below a certain threshold to enable normal DV pattern formation. This may provide the system with robustness against fluctuating BMP signaling levels. In our theoretical analysis, the homogeneous steady state of Sog and BMP undergoes a Turing instability when NF-κB/Dorsal-dependent activation of *sog* reaches a threshold (*Figure 9*). Then, rapid diffusion of BMP-Sog complexes and derepression of *sog* upon removal of BMP from the ventral side lead to the formation of a stripe with high expression of *sog*. The position of the stripe is determined by the initial NF-κB/Dorsal polarity cue.

## BMP patterns the entire DV axis in *Oncopeltus*

In all insects studied so far the specification of mesodermal and mesectodermal cell fates requires Toll signaling. In *Drosophila* and *Tribolium*, the shape of the NF-κB/Dorsal gradient directly or indirectly determines the width of the mesodermal domain (*Chen et al., 2000*; *Hong et al., 2008*; *Nunes da Fonseca et al., 2008*). In *Nasonia* all ventrally expressed genes (e.g., *twi*) are first turned on in a narrow stripe straddling the ventral midline (*Özüak et al., 2014b*). This region also shows high levels of *cact* expression indicating high activity of Toll signaling. Subsequently, *cact* expression disappears, however, the expression of ventral genes expands. The size of their final domains is determined by repressive BMP signaling from the dorsal side, since KD of BMP leads to progressive expansion of ventral gene expression resulting in a massive, albeit nonuniform expansion of the mesoderm. Thus, in *Nasonia* BMP effects all subdivisions of the DV axis (*Özüak et al., 2014b*). However, the expansion of ventral genes remains dependent on their prior activation by Toll, as *Toll bmp* double KD embryos lack the expression of ventral genes like *twi*.

In *Oncopeltus dpp* KD embryos, *twi* is completely derepressed, that is, its expression is uniform around the embryonic circumference and the developing embryos are fully mesodermalized (*Figures 2E, 4F*). This phenotype does not result from a progressive expansion of a narrow domain as in *Nasonia*. Most importantly, the same phenotype is observed in *Toll dpp* double KD embryos, with the exception that *twi* expression in addition expands anteriorly due to an AP function of Toll signaling (*Figure 5L*). These data suggest that Toll signaling in *Oncopeltus* is no longer strictly required to activate ventral genes. As a consequence, all cell fate decisions along the DV axis of *Oncopeltus* ultimately depend on different levels of BMP signaling.

As pointed out previously, the early pattern of BMP signaling in *Oncopeltus* seems to be very simple with a plateau of high signaling at the dorsal side and a broad domain lacking BMP signaling ventrally (*Figure 3—figure supplement 1*). Accordingly, the early DV fate map of *Oncopeltus* has apparently only few subdivisions. Although we have thus far not identified genes expressed specifically on the dorsal side, we expect such genes to have broad expression domains encompassing 30–50% of the embryonic circumference. Likewise, the ventral *twi* and early *sim* domains are broad (data not shown). *sim* refines to lateral stripes bordering *twi* during later blastoderm stages. None of the columnar genes (*vnd*, *ind*, *msh*) show stripe-like expression in lateral regions of the blastoderm as in holometabolous insects (*von Ohlen and Doe, 2000*; *Wheeler et al., 2005*; *Buchta et al., 2013*). We therefore expect that the early BMP gradient provides little patterning

information on either the dorsal or ventral side, and that further refinement occurs progressively during and after gastrulation. This refinement also includes large-scale morphogenetic movements. For example, the narrow *twi* domain of germ band embryos (*Figure 4C*) results from massive convergent extension during anatrepsis (posterior invagination of the embryo and the amnion into the yolk [*Panfilio, 2008*]).

## In *Oncopeltus* Toll function is restricted to polarizing BMP signaling

Several lines of evidence suggest that the Toll gradient acts differently in *Oncopeltus* compared to the known holometabolous insects: (i) BMP is epistatic to Toll, (ii) ventrally expressed genes can be readily suppressed by increased BMP signaling, (iii) the repression of *sog* by BMP makes the distribution of BMP signaling largely independent from that of Toll signaling. Although we have not demonstrated the latter point experimentally, modeling shows that adding the repression of *sog* by BMP to the well known reaction diffusion system (*Eldar et al., 2002*) leads to a decoupling of input (Toll signaling) and output (BMP signaling) patterns (*Figures 9, 11*). Toll remains essential at the ventral side to initiate patterning, and therefore, it would be highly interesting to monitor Toll activity by looking at the NF-κB/Dorsal distribution.

In absence of functional antibodies we used the expression of the early Toll target gene *cact* as a proxy for Toll signaling. In all known holometabolous insects *cact* is activated by the NF-κB/Dorsal gradient. This applies even to *Drosophila* as demonstrated with the help of an enhancer reporter construct which shows *twist*-like expression (*Sandmann et al., 2007*). However, due to maternal loading of *cact,* its zygotic expression apparently has little functional relevance in *Drosophila* (*Roth et al., 1991*). In *Tribolium*, *cact* is only zygotically expressed and tightly follows the shifting gradient of NF-κB/Dorsal (*Nunes da Fonseca et al., 2008*). Finally, in *Nasonia*, *cact* expression is restricted to a narrow stripe straddling the ventral midline where the expression of all other Toll signaling-dependent ventral genes is initiated (*Özüak et al., 2014b*). Although the NF-κB/Dorsal distribution is not known from *Nasonia*, a cluster of NF-κB/Dorsal binding sites in the vicinity of the *cact* transcript suggests a direct regulatory input by Toll signaling. By analogy we assume that the broad, weakly graded expression of *cact* in *Oncopeltus* reflects a flat Toll signaling gradient which extends from the ventral to the dorsal half of the embryo (*Figure 7*).

The upstream cascade, which leads to Toll activation in *Drosophila* (*Stein and Stevens, 2014*) appears to be largely conserved in *Oncopeltus* (data not shown). Preliminary data suggest that the asymmetry of Toll signaling originates from asymmetric eggshell cues that are established during oogenesis. By activating Toll signaling in a broad gradient, these eggshell cues would provide global polarity to the embryo, which is essential for establishing bilateral symmetry. Such a strong geometric influence does not exclude, but rather enables, certain forms of self-organized patterning. Classical insect embryology had described many instances of partial or complete twinning after experimental interference with patterning along the DV axis of the egg (*Sander, 1976*). These experiments were not restricted to hemimetabolous insects, but included examples from beetles as well as butterflies.

The most famous set of experiments was conducted by Sander with *Euscelis*, a leaf hopper which (like *Oncopeltus*) belongs to the Hemiptera (*Sander, 1971*). Sander produced not only left and right, but also dorsal and ventral egg fragments and was able to recover apparently complete germ rudiments from all fragments. These findings could not previously be explained on the basis of the fairly deterministic mechanism of DV axis formation known from *Drosophila*. The mechanism we have discovered in *Oncopeltus* can, in principle, account for this type of axis duplication (*Box 1*). As long as Toll activity is globally provided so that *sog* can also be activated in the dorsal half of the egg, a new round of patterning can be initiated dorsally. A prerequisite for pattern re-initiation is a diffusion barrier, which prevents the transport of (inhibitory) BMP molecules from the ventral to the dorsal side. The guillotine-like mechanism with which Sander separated the egg fragments provided such a barrier.

In summary, the data presented here not only provide a potential explanation for experiments from classical insect embryology, they also suggest a scenario of how the elaborate morphogen function of Toll signaling found in *Drosophila* could have originated and evolved (*Figure 1*). Ancestrally, Toll signaling might have only provided a polarizing function for a self-organizing BMP system responsible for patterning the entire DV axis. Within certain lineages (e.g., flies, beetles and wasps) Toll signaling became more important in directly specifying cell fates along the axis, gradually replacing ancestral BMP functions.

Our data might also help to explain the transition from the ancestral immune function of Toll found in most metazoan lineages (*Gilmore and Wolenski, 2012*) to its unique role in DV patterning restricted to insects. Recent findings show that insect eggs are immune competent (*Jacobs and van der Zee, 2013*; *Jacobs et al., 2014*). We suggest that this also applied to ancestral insects and that furnishing their eggs with a Toll-mediated pathogen defense system was crucial for early insects to adopt a terrestrial life style. The activation of Toll signaling by eggshell cues might have been fairly uniform throughout the embryo. Subsequently, only a mild polarization of Toll signaling together with weak transcriptional inputs on *sog* were sufficient to initiate the co-option of Toll signaling for DV patterning.

# Materials and methods

## Gene annotation and analysis

Putative *O. fasciatus* homologs of Toll and BMP signaling components were found by a local blast against a maternal and embryonic transcriptome (*Ewen-Campen et al., 2011*) using the NCBI blast+ toolkit and BioEdit software, or by degenerate PCR followed by RACE PCR using the SMARTER RACE kit (Clontech, France) to extend the sequence information. Specific primers of all candidates were designed for sequencing, cloning and to confirm the homology with *Drosophila* and *Tribolium* (*Supplementary file 1*). Phylogenetic and molecular evolutionary analyses were then conducted using MEGA version 5 (*Tamura et al., 2011*) or phylogeny.fr (*Dereeper et al., 2008*). All *Oncopeltus* gene sequences have been submitted to GenBank.

## RNA interference and KD efficiency validation

To knock down gene function, gene-specific double-stranded RNA (dsRNA) (0.1–8 µg/µl) for parental RNAi was prepared as previously described and injected into virgin females (*Nunes da Fonseca et al., 2008*). After injection, embryos were collected, fixed and stored in methanol at −20°C as previously described (*Liu and Kaufman, 2004*) for further phenotype analysis. Total RNA from a single cohort of staged embryos was homogenized and extracted by TRIzol reagent (Life Technologies, Germany) with DNase treatment, and cDNA was synthesized with the VILO Kit (Invitrogen, Germany), following manufacturers' protocols. Gene expression analysis using semi-quantitative RT-PCR was performed using gene-specific primers, with an annealing temperature of 60°C, and 30 thermocycles.

## In situ hybridization and immunohistochemistry

Detection of gene expression was performed by in situ hybridization (ISH) with digoxigenin-labeled probes as previously described (*Liu and Kaufman, 2004*). The double ISH was performed with digoxigenin (DIG) and biotin labeled probes hybridized simultaneously followed by incubation with anti-Biotin-AP (1:5000, Roche, Germany). After the first round of staining the anti-Biotin-AP antibody was inactivated by treating the embryos with 0.1 M Glycine-HCl, pH = 2.2, 0.1% Tween20 for 10 min, followed by washing, blocking and incubation with the second AP antibody (anti-DIG-AP, 1:5000, Roche, Germany). For color reactions we used the HNPP Fluorescent Detection Set (Roche, Germany) and NBT/BCIP. Immunostaining was performed using anti-Phospho-Smad1/5 (41D10) rabbit antibodies (Cell Signaling, Germany) with 1:30 dilution. We introduced the TSA plus DNP system (Perkin Elmer, Waltham, MA) to amplify the signal before DAB detection (or DAB with nickel ammonium sulfate).

## Reaction-diffusion model of the BMP/Sog system

### Qualitative description of the model

We build a reaction-diffusion model of the BMP/Sog system in *Oncopeltus* based on a minimal number of components and their interactions. At the core are the following processes:

- Expression of BMP and of *sog*. The expression of *sog* is repressed by BMP and (indirectly) enhanced by NF-κB/Dorsal, whereas BMP is constitutively expressed.
- Binding of Sog protein to BMP protein to form a Sog-BMP complex, and the dissociation of the Sog-BMP complex.
- Diffusion of BMP, Sog, and the Sog-BMP complex. Crucially, the diffusion constant of the Sog-BMP complex is higher than that of either BMP or Sog.
- The presence of an initial distribution of NF-κB/Dorsal, leading to a locally enhanced expression of *sog*. This initial distribution acts as a polarity cue.
- Degradation of BMP, Sog, and NF-κB/Dorsal. Degradation of NF-κB/Dorsal makes this polarity cue to disappear with time.

Before formulating and analyzing this model we discuss its dynamics qualitatively. Consider a small 'seed' of elevated Sog concentration in a specific location (for instance as the result of the polarity cue provided by NF-κB/Dorsal). The Sog molecules bind BMP and the complexes diffuse away quickly, leading to a depletion of BMP. Since BMP represses Sog, this leads to a local increase in Sog production, causing the small 'seed' to grow. The steady-state consists of areas where BMP has been depleted (and where levels of Sog are high), bordering areas with high levels of BMP (and low levels of Sog). In the former, *sog* is not repressed due to the absence of BMP, and BMP is absent due to complex formation with Sog and diffusive transport. In the latter, high levels of BMP repress *sog* expression. The resulting non-equilibrium steady state will be examined in detail elsewhere.

## Comparison with other models

We consider only one BMP ligand and neglect the fact that BMPs are secreted as homodimers or heterodimers (*Shimmi et al., 2005*), although we have experimental evidence for a second BMP ligand in *Oncopeltus* (*Figure 2—figure supplement 1*). This is in agreement with other recent theoretical models for BMP gradient formation (*Mizutani et al., 2005*; *Shimmi et al., 2005*; *Ben-Zvi et al., 2008*; *Umulis et al., 2010*; *Peluso et al., 2011*; *Inomata et al., 2013*) and reflects the notion that additional ligands might contribute to increased robustness (*Shimmi et al., 2005*), but have no essential impact on the mechanisms of pattern formation. The binding of the BMPs to their receptors is not part of our model although we are aware of receptor-mediated degradation affecting the mean free path of the ligand (*Mizutani et al., 2005*). The model also neglects the binding of Tsg to Sog-BMP complexes and does not explicitly mention Tolloid (Tld), the enzyme that cleaves Sog. This can be justified by the fact that in blastoderm embryos *tsg* transcripts are not detectable (by ISH) suggesting very weak uniform expression and that *tld* is evenly expressed around the embryonic circumference (*Figure 2—figure supplement 1*). Similar assumptions have been made in a recent model for BMP signaling in *Xenopus* (*Inomata et al., 2013*).

Our minimal model has similarity to one originally suggested by *Eldar et al. (2002)* (*Meinhardt and Roth, 2002*) with the crucial difference that *sog* expression itself is controlled by BMP signaling. The NF-κB/Dorsal gradient in our model plays a similar role as the source density in the Gierer-Meinhardt model, which was used to explain regulative behavior in hydra (*Gierer and Meinhardt, 1972*). In fact the mechanism of pattern formation bears some similarity with the local activation-lateral inhibition mechanism proposed by *Gierer and Meinhardt (1972)*: BMP inhibits Sog production through transcriptional repression. However, this is the sole regulatory interaction in our model. Local self-activation arises from the fast transport of BMP within Sog-BMP complexes, which moves the repressor for *sog* from regions of high *sog* expression (increasing *sog* expression in those regions) to regions of low *sog* expression (decreasing *sog* expression there).

## Reaction-diffusion model

To describe the repression of *sog* by BMP we use a Michaelis–Menten model; the rate of *sog* expression is proportional to $1/(1 + b/b_0)$ where $b$ is the concentration of BMP and $b_0$ is the concentration of BMP reducing the expression of *sog* by a factor of two. The constant of proportionality depends on the NF-κB/Dorsal concentration $d$, which enhances *sog* expression. In the absence of BMP, the *sog* expression rate $(\eta_0 + \eta_1 d/d_0)/(1 + d/d_0)$ extrapolates between $\eta_0$ at $d = 0$ and $\eta_1 > \eta_0$ at high levels of $d$. $d_0$ denotes the concentration where the intermediate transcription rate $(\eta_0 + \eta_1)/2$ is reached. Combining the repression of *sog* by BMP with its enhancement by NF-κB/Dorsal we obtain the transcription rate of *sog* as a function of the concentrations of BMP and NF-κB/Dorsal as,

$$\eta_s(b, d) = \frac{\eta_0 + \eta_1 \, d/d_0}{(1 + b/b_0)(1 + d/d_0)}. \tag{1}$$

This model of expression regulation can also be derived from a thermodynamic model of two transcription factors (a repressor and an activator) independently binding to a regulatory region.

We model the geometry of the embryo as a cylinder with circumference $l_x$. When taking all concentrations to be constant along the axis of the cylinder, one obtains an effectively one-dimensional model. This assumption will be examined below, where the dynamics on the surface of the cylinder are considered. In the one-dimensional case, the concentrations $s$ of Sog, $b$ of BMP, $c$ of the Sog-BMP complex, and $d$ of the transcription factor NF-κB/Dorsal depend on time and on the

variable $x \in [0, l_x]$ running along the circumference of the cylinder, with $x = 0$ denoting the ventral side and $x = lx/2$ the dorsal side.

We now formulate a reaction-diffusion model for the concentrations $s(x,t)$ of Sog, $b(x,t)$ of BMP, $c(x,t)$ of the Sog-BMP complex, and $d(x,t)$ of the transcription factor NF-κB/Dorsal. Under the processes described above, these concentrations evolve as,

$$\partial_t s = D_s \nabla^2 s + \eta_s(b, d) - k_+ sb + k_- c - \alpha_s s,$$

$$\partial_t b = D_b \nabla^2 b + \eta_b - k_+ sb + k_- c + \alpha_s c - \alpha_b b,$$

$$\partial_t c = D_c \nabla^2 c + k_+ sb - k_- c - \alpha_s c,$$

$$\partial_t d = -\alpha_d d.$$

(2)

In one dimension $\nabla^2 = \partial^2/\partial x^2$, $D$ with appropriate subscript denotes the diffusion constants, analogously $\eta$ the rates of gene expression, $\alpha$ the degradation rates, and $k_+$ and $k_-$ the binding and unbinding rates of Sog and BMP. *Table 1* gives the parameters used here. We neglect degradation of BMP in the complex, although our results do not depend on this. It turns out that many of these parameters can be changed over at least one order of magnitude without affecting pattern formation (*Table 2*).

Writing the *sog* expression rate as $\eta_s(b, d) \equiv \bar{\eta}_s(d)/(1 + b/b_0)$ with $\bar{\eta}_s(d) \equiv \frac{\eta_0 + \eta_1 d/d_0}{1 + d/d_0}$, we see that the NF-κB/Dorsal concentration $d$ controls the difference in *sog* expression between high and low levels of BMP. We will show below that pattern formation crucially depends on (i) a sufficiently high diffusion rate of the Sog-BMP complex and (ii) repression of *sog* expression by BMP. The homogeneous state (uniform concentrations of BMP, Sog, and the Sog-BMP complex) can be stable at low levels of NF-κB/Dorsal, but becomes unstable (via the mechanism above) once a certain critical NF-κB/Dorsal level is reached. Once a stripe has formed, however, it can persist even in the absence of NF-κB/Dorsal (*Figure 8*).

We now explore the dynamics of pattern formation in this model starting from different concentrations of the transcription factor NF-κB/Dorsal (shown in gray), resulting in steady-state concentrations of Sog and BMP shown blue and red, respectively. Starting from small levels of NF-κB/Dorsal, a steady state with uniformly low level of Sog arises (*Figure 9*, left). A threshold amount of NF-κB/Dorsal is required initially to form a stripe of high Sog concentration (*Figure 9*, center). As a result, small fluctuations in the NF-κB/Dorsal concentration thus do not lead to the formation of a stripe. The size of this stripe does not change if the initial amount of NF-κB/Dorsal is increased (*Figure 9*, right). This NF-κB/Dorsal-induced instability of the homogeneous state is also behind the twinning phenomenon (*Box 1*): if the level of NF-κB/Dorsal exceeds the critical threshold everywhere in the system, a stripe of Sog centered on the maximum of NF-κB/Dorsal forms. Elsewhere, *sog* is repressed by BMP and transported away from the high-Sog stripe via the Sog-BMP complex. If the system is divided into two separate halves, this transport is interrupted, but the NF-κB/Dorsal level is still above the critical threshold everywhere. Now there are two maxima of NF-κB/Dorsal (one in each of the two halves), and a stripe of Sog forms at each of them.

## Stability analysis

We perform a linear stability analysis to determine when a spatially homogeneous steady-state solution is instable against small sinusoidal spatial oscillations. For now, we consider a spatially uniform NF-κB/Dorsal concentration $d$ as a parameter that can be tuned to take on

**Table 2**. Range of model parameter values where a single stripe is formed

| | |
|---|---|
| $\bar{\eta}_s$ | $6 \times 10^{-4}$–$1.4 \times 10^{-3}$ |
| $b_0$ | 0.1–0.5 |
| $\eta_b$ | $8 \times 10^{-6}$–$10^{-4}$ |
| $\alpha_s$ | $1.6 \times 10^{-3}$–$1.2 \times 10^{-2}$ |
| $\alpha_b$ | $0$–$10^{-4}$ |
| $K_+$ | 0.05–200 |
| $k_-$ | 0–0.5 |
| $D_s$ | $0$–$10^{-9}$ |
| $D_b$ | $0$–$10^{-11}$ |
| $D_c$ | $7 \times 10^{-10}$–$10^{-6}$ |

Each parameter is varied keeping the other parameters fixed at the values specified in *Table 1*. One exception is the parameters $\eta_0$ and $\eta_1$, which affect pattern formation jointly through the parameter $\bar{\eta}_s(d) \equiv \frac{\eta_0 + \eta_1 d/d_0}{1 + d/d_0}$ which is set to $1.2 \times 10^{-3}$ (except in the first line, where this parameter itself is varied).

different values. Then the reaction-diffusion *Equations (2)* become,

$$\partial_t s = D_s \nabla^2 s + f_s(s(x,t), b(x,t), c(x,t)),$$

$$\partial_t b = D_b \nabla^2 b + f_b(s(x,t), b(x,t), c(x,t)),$$

$$\partial_t c = D_c \nabla^2 c + f_c(s(x,t), b(x,t), c(x,t)),$$

(3)

with the shorthands,

$$f_s(s,b,c) = \bar{\eta}_s(d)/(1 + b/b_0) - k_+ sb + k_- c - \alpha_s s,$$

$$f_b(s,b,c) = \eta_b - k_+ sb + k_- c + \alpha_s c - \alpha_b b,$$

(4)

$$f_c(s,b,c) = k_+ sb - k_- c - \alpha_s c.$$

We now consider a spatially homogeneous fixed point of *Equation (3)* defined by $s(x,t) = \bar{s}$, $b(x,t) = \bar{b}$, $c(x,t) = \bar{c}$ with $0 = f_s(\bar{s}, \bar{b}, \bar{c}) = f_b(\bar{s}, \bar{b}, \bar{c}) = f_c(\bar{s}, \bar{b}, \bar{c})$. In the vicinity of such a fixed point the reaction-diffusion dynamics can be written as,

$$\partial_t \begin{pmatrix} s(x,t) \\ b(x,t) \\ c(x,t) \end{pmatrix} = \mathbf{D} \frac{\partial^2}{\partial x^2} \begin{pmatrix} s(x,t) \\ b(x,t) \\ c(x,t) \end{pmatrix} + \mathbf{A} \begin{pmatrix} s(x,t) - \bar{s} \\ b(x,t) - \bar{b} \\ c(x,t) - \bar{c} \end{pmatrix},$$

(5)

where **A** is a matrix of partial derivatives evaluated at the fixed point,

$$\mathbf{A} = \begin{pmatrix} \frac{\partial f_s}{\partial s}\big|_0 & \frac{\partial f_s}{\partial b}\big|_0 & \frac{\partial f_s}{\partial c}\big|_0 \\[2mm] \frac{\partial f_b}{\partial s}\big|_0 & \frac{\partial f_b}{\partial b}\big|_0 & \frac{\partial f_b}{\partial c}\big|_0 \\[2mm] \frac{\partial f_c}{\partial s}\big|_0 & \frac{\partial f_c}{\partial b}\big|_0 & \frac{\partial f_c}{\partial c}\big|_0 \end{pmatrix}$$

(6)

and **D** is the diagonal matrix,

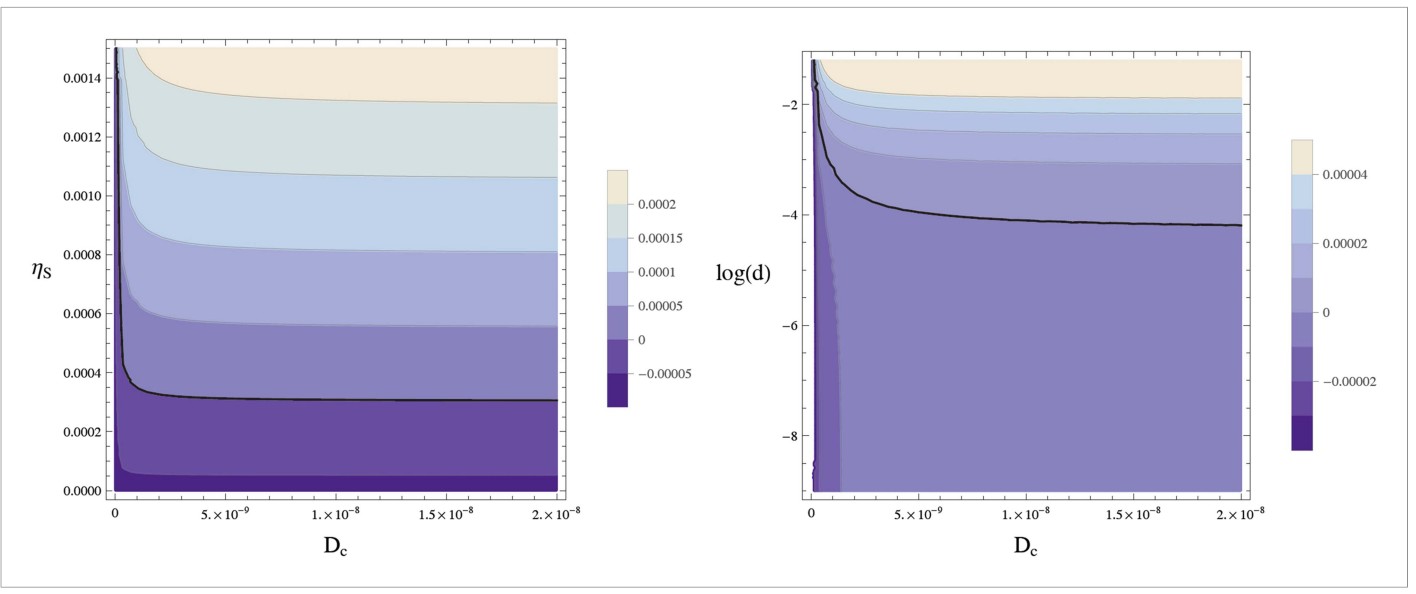

**Figure 12**. Stability of the homogeneous fixed point. This contour plot shows the largest eigenvalue $w$ of $\mathbf{A} - \mathbf{D}k^2$ for $k = 2\pi/l_x$. The thick line separates the parameters leading to a stable homogeneous fixed point ($w < 0$) from an instable homogeneous fixed point ($w > 0$). (left) $w$ is plotted as a function of the diffusion constant of the Sog-BMP complex and the rate of *sog* expression at zero BMP, $\bar{\eta}_s$. (right) The same data are plotted against $\log(d)$ using $\bar{\eta}_s(d) \equiv \frac{\eta_0 + \eta_1 d/d_0}{1 + d/d_0}$. The homogeneous fixed point becomes unstable for sufficiently large values of the diffusion constant of the complex $D_c$ and the concentration of NF-κB/Dorsal $d$. The remaining parameters are as given in *Table 1*.

$$\mathbf{D} = \begin{pmatrix} D_s & & \\ & D_s & \\ & & D_c \end{pmatrix}.$$ (7)

The standard ansatz to check stability of the fixed point against small spatial oscillations proceeds by adding a sinusoidal term to the fixed point (*Turing, 1952*).

$$\begin{pmatrix} s(x, t) \\ b(x, t) \\ c(x, t) \end{pmatrix} = \begin{pmatrix} \bar{s} \\ \bar{b} \\ \bar{c} \end{pmatrix} + e^{wt+ikx} \begin{pmatrix} \delta s \\ \delta b \\ \delta c \end{pmatrix}.$$ (8)

Inserting this ansatz into *Equation (5)* gives the eigenvalue equation,

$$w \begin{pmatrix} \delta s \\ \delta b \\ \delta c \end{pmatrix} = (\mathbf{A} - \mathbf{D}k^2) \begin{pmatrix} \delta s \\ \delta b \\ \delta c \end{pmatrix}.$$ (9)

The fixed point is stable against small spatial sinusoidal perturbations in one spatial dimension if and only if for all values of the wave vector $k$ compatible with the cylindrical geometry all eigenvalues $w$ of the matrix $\mathbf{A} - \mathbf{D}k^2$ have a negative real part. *Figure 12* shows the largest eigenvalue of $\mathbf{A} - \mathbf{D}k^2$ for the wavenumber $k = 2\pi/l_x$, that is, the smallest non-zero wavenumber compatible with the circular geometry, showing how the homogeneous steady state becomes unstable for sufficiently large values of $D_c$ and $d$.

## Dynamics in two dimensions

We now explore the dynamics of the model constructed above on a two-dimensional cylinder. The surface of the cylinder is described by a variable $y \in [0, l_y = 0.002]$ running along its axis, and $x \in [0, l_x]$ running along the circumference. The equations of motion follow from *Equation (2)* with $\nabla^2 = \partial^2/\partial x^2 + \partial^2/\partial y^2$ and open boundary conditions at the two ends of the cylinder. We use the same parameters as in the one-dimensional case (*Table 1*). Starting from an initial distribution of NF-κB/Dorsal along the ventral side of the cylinder, a stripe of Sog forms along the cylinder (*Figure 10*). All concentrations turn out to be independent of $y$. We find this also holds in the steady state if the initial distribution of NF-κB/Dorsal varies in the $y$-direction. This becomes more apparent in *Figure 11*, where the initial distribution of NF-κB/Dorsal from *Figure 10* is shown in a contour plot, alongside the steady-state concentration of Sog. Thus, our model produces stripes of constant width (*sog* expression), which are centered on the ventral midline defined by NF-κB/Dorsal. This is a remarkable feature as it had been difficult to produce striped patterns centered on the midline with local activation-lateral inhibition or substrate depletion mechanisms (*Meinhardt, 2004*).

## Acknowledgements

This article is dedicated to the memory of the late Klaus Sander (1929-2015), who discovered anteriorposterior morphogen gradients in insects and the regulative patterning along the DV axis. His experiments provided the main motivation for the research conducted in SR's lab, leading to the current paper. We thank Rodrigo Nunes da Fonseca for comments and discussions and are grateful to Waldemar Wojciech and Matt Benton for suggestions, corrections and help with *Figure 1*. This work has been supported by the CRC 680 of the DFG and the NRW International Graduate School of Development Health and Disease.

## Additional information

### Funding

| Funder | Grant reference | Author |
| --- | --- | --- |
| Deutsche Forschungsgemeinschaft (DFG) | CRC 680 | Yen-Ta Chen, Jeremy A Lynch, Kristen A Panfilio, Michael Lässig, Johannes Berg, Siegfried Roth |

| Funder | Grant reference | Author |
|---|---|---|
| Ministerium für Innovation, Wissenschaft und Forschung des Landes Nordrhein-Westfalen | NRW International Graduate School of Development Health and Disease | Yen-Ta Chen |

The funders had no role in study design, data collection and interpretation, or the decision to submit the work for publication.

## Author contributions

LS, Y-TC, Conception and design, Acquisition of data, Analysis and interpretation of data, Drafting or revising the article; AD, Acquisition of data, Analysis and interpretation of data; JAL, KAP, ML, JB, SR, Conception and design, Analysis and interpretation of data, Drafting or revising the article

## Additional files

### Supplementary file

• Supplementary file 1. PCR primers for production of ISH probes and dsRNA.

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
