## [Decision Letter]

Thank you for sending your work entitled “Toll's patterning role emerged as a polarity cue for self-regulatory BMP signaling” for consideration at *eLife*. Your article has been favorably evaluated by Diethard Tautz (Senior editor), a Reviewing editor, and three reviewers.

Sachs et al., study DV patterning in the milkweed bug *Oncopeltus fasciatus*. They use RNAi methods and mathematical modeling to study interactions between the Toll and BMP signaling pathways. From their RNAi data, the authors conclude that (1) BMP signaling is active throughout the DV axis and must be repressed ventrally through expression of *sog* to allow ventral patterning; and (2) Toll is not required to support any cell types but is required for symmetry breaking.

All reviewers agreed that this is an interesting study addressing an important question in developmental biology, namely how do DV patterning mechanisms in different animals relate and what was the ancestral system like. The authors make a convincing argument for looking at the milkweed bug.

However, it was also agreed by all that more information is needed, especially in view of the recent CB paper from the PI lab on *Nasonia,* which already makes the point regarding the decreasing role of the Toll pathway. Several additional data are therefore essential:

1) More information on the expression patterns of the different players in the BMP pathway is essential: Where is *dpp* expressed? It is implied that *dpp* is no longer under Toll-dependent regulation in the milkweed bug (e.g. Figure 1 diagram), however no in situ data is included. The authors also suggest that multiple BMP ligands may be supporting activation of this signaling pathway. Their expression should be shown and discussed. Could they be repressed by Dorsal?

2) What is the expression pattern of Tolloid? Similar arguments to point 1.

3) The ability to trace the expression of the Zen homologue as an indicator of maximal BMP signaling in different backgrounds could be very informative.

4) More information on the pattern of the Toll gradient shape would be important.

5) Additional support for the *Toll1-dpp-*RNAi results should be provided. (one may worry that the *Toll1* RNAi support weak activation that becomes apparent in the absence of BMP signaling; What does *Toll1-dl1-dpp* RNAi look like? How about *Toll1-tld* RNAi?).

In addition, it would be interesting to directly define the shape of the Dorsal gradient (e.g. by indirect immunofluorescence if antibodies are available or could be produced in a reasonable length of time) as this shape is an important output of the modeling. In Box 1, the first diagram suggests that the height and width of the Dorsal gradient increases in time. What is the evidence to support this in the milkweed bug? In Box 1, the second diagram shows that in the middle of the “twinned” embryo, a peak of BMP signaling abuts a domain of high *sog* expression. How is this possible?

The authors should also discuss how *sog* can be expressed in a small ventral domain in dl1-RNAi mutants. There must be some “polarity cue” and yet development does not proceed normally. So the levels of Dorsal are important to some degree? Previous studies in *Drosophila* have provided evidence that genes expressed in lateral regions (e.g. brinker) are ubiquitously expressed in *dl dpp* double mutants. Is *sog* expressed ubiquitously within *dl dpp* double *Drosophila* mutants?

Reviewer 2:

In the manuscript by Sachs et al., the authors present an analysis of DV patterning in the milkweed bug *Oncopeltus fasciatus*. They use RNAi methods and mathematical modeling to study interactions between the Toll and BMP signaling pathways. From their RNAi data, the authors conclude that (1) BMP signaling is active throughout the DV axis and must be repressed ventrally through expression of *sog* to allow ventral patterning; and (2) Toll is not required to support any cell types but is required for symmetry breaking. This is an interesting study addressing an important question in developmental biology, namely how do DV patterning mechanisms in different animals relate and what was the ancestral system like. The authors make a convincing argument for looking at the milkweed bug.

The data are suggestive of the hypothesis, that Toll signaling is symmetry breaking, but do not demonstrate it. Alternately, rather, the data show that BMP signaling acts broadly and then is refined.

Furthermore, the addition of some data and controls is necessary. Methods available in this new model system are limited, understandably. Perhaps this is why the mathematical modeling was included, because it was accessible; and yet I found the addition of modeling to explain “twinning” more confusing than helpful. In any case, the individual and joint contributions of dorsal genes, BMP ligand expression and roles, and the possibility of off-target effects for RNAi constructs should (and can) be addressed.

In summary, this is a very interesting study and the results are significant. However, either additional experimental evidence must be included or the title should be revised. If the study is revised to refocus on the broad role of BMP signaling, then the novelty of the study is perhaps compromised by the authors' recent study in *Nasonia* (Ozuak et al., 2014).

Specific comments:

1) Where is *dpp* expressed? It is implied that *dpp* is no longer under Toll-dependent regulation in the milkweed bug (e.g. Figure 1 diagram), however no in situ data is included. The authors also suggest that multiple BMP ligands may be supporting activation of this signaling pathway. Their expression should be shown and discussed. Could they be repressed by Dorsal?

2) Previous studies in *Drosophila* have provided evidence that genes expressed in lateral regions (e.g. brinker) are ubiquitously expressed in *dl dpp* double mutants. Is *sog* expressed ubiquitously within *dl dpp* double *Drosophila* mutants?

3) In Box 1, the first diagram suggests that the height and width of the Dorsal gradient increases in time. What is the evidence to support this in the milkweed bug?

4) In Box 1, the second diagram shows that in the middle of the “twinned” embryo, a peak of BMP signaling abuts a domain of high *sog* expression. How is this possible?

Reviewer 3:

The paper by Sachs et al. examines the roles of the Toll and BMP pathways in DV patterning of the Hemimetabolous insect *Oncopelus*. With the universal role of BMP DV patterning in multicellular organisms, and the restricted role of the Toll pathway in DV patterning of Holometabolous insects like *Drosophila*, the question was what is the relation of the Toll and BMP pathways in DV patterning of *Oncopelus*. They show that BMP signaling is required to repress expression of early ventral and lateral genes (*twi* and *sog*) in the dorsal region. At early embryogenesis Sog displays a low and uniform expression. The role of Toll signaling is only to break this symmetry and lead to an increase (even a moderate one) in Sog levels at the ventral side. In a double RNAi mutant for Toll and *dpp*, ventral fates take over the entire embryo, indicating that in this species this is the default, and the role of Toll is only to repress BMP signaling in the ventral region by elevating the level of Sog. Because of the cross regulatory interactions between *dpp* and Sog, this leads to a stable BMP activation pattern and expression of the cardinal genes in restricted domains. Computational analysis indicates that this regulatory circuitry is robust to fluctuations, and requires only a minimal bias by the Toll pathway.

In general, the detailed dissection of developmental pathways in organisms that are removed from *Drosophila*, which is successfully used by the Roth lab, and especially the deviations from what we know in *Drosophila*, provide a fresh view not only on the evolution of these pathways, but also on their regulatory logic. This work is a case in point.

Several experiments were missing and would provide a broader view of the topic:

1) What is the expression profile of *dpp* in *Oncopelus* embryos at different stages? This is important because it would provide the context to interpret the results in Sog RNAi embryos. Is the global activation of the BMP pathway a result of BMP diffusion from the dorsal region, or a reflection of a uniform expression of *dpp*? If *dpp* expression is restricted to the dorsal region, is the repression of expression in the ventral region dependent on Toll signaling, similar to the induction of Sog?

2) What is the expression pattern of Tolloid? Similar arguments to point 1.

3) What is the nuclear distribution pattern of Dorsal1 and Dorsal2 in wt embryos, and in embryos in which *dpp* signaling has been eliminated? I know that this analysis would require a significant work of raising antibodies, but perhaps it could be circumvented by injection of a plasmid expressing tagged or GFP-linked Dorsal1. It would be important to see the pattern of the Toll activation gradient, and to assess if it extends sufficiently into the dorsal side, to account for the proposed bias in that region in the that is relevant to the Sander experiment (Box 1). It would also be important to demonstrate that the Toll gradient is the primary symmetry-breaking event, and is not affected by the loss of the downstream BMP pathway.

Several points should be clarified further in the text, without a requirement for additional experiments:

Sog levels were not erased but only reduced in the Sog RNAi embryos (Figure 2). While the residual Sog is now uniformly expressed, there still appears to be some difference between the level of pMad in the dorsal vs. ventral parts of the embryo (Figure 3). This would not agree with the model, but perhaps the embryos shown are not representative, or maybe there is still some bias in the expression of Sog which resolves only to a very shallow pMad pattern because of the limited levels of Sog?

The central point of the paper is the switch from a uniform distribution of Sog, to the restricted expression of Sog in the ventral and ventro-lateral domains, and the complementary pattern of BMP activation, triggered by the ventral bias of Toll signaling. The ability to convert even a small bias (as in the middle panel of Figure 8) to a bistable robust BMP pattern, clearly relies on transport of the Sog-BMP complex and its cleavage in the dorsal region by Tolloid. In this respect the logic of modulating BMP signaling is similar to what was described in *Drosophila*. It is not clear however whether this process is also responsible (as it is in *Drosophila*) for generating a graded pattern WITHIN the domain of BMP signaling. As far as I could tell, the pMad levels in embryos that are depleted for Sog were comparable to wt, suggesting that in the absence of inhibition by Sog, the levels of *dpp* are sufficient to elicit maximal signaling and no further concentration of the ligand is required. Also, the late patterns of pMad show sharp borders and fairly uniform expression within the dorsal region (Figure 3). I would like to see some more discussion of this issue, regarding the question whether a graded BMP activation pattern is or is not generated within the dorsal region.

---

## [Author Response]

*Sachs et al., study DV patterning in the milkweed bug* Oncopeltus fasciatus*. They use RNAi methods and mathematical modeling to study interactions between the Toll and BMP signaling pathways. From their RNAi data, the authors conclude that (1) BMP signaling is active throughout the DV axis and must be repressed ventrally through expression of* sog *to allow ventral patterning; and (2) Toll is not required to support any cell types but is required for symmetry breaking*.

*All reviewers agreed that this is an interesting study addressing an important question in developmental biology, namely how do DV patterning mechanisms in different animals relate and what was the ancestral system like. The authors make a convincing argument for looking at the milkweed bug*.

*However, it was also agreed by all that more information is needed, especially in view of the recent CB paper from the PI lab on* Nasonia, *which already makes the point regarding the decreasing role of the Toll pathway. Several additional data are therefore essential*:

The reviewers have noted that a decreasing role of Toll signaling accompanied by an expanded function of BMP was already described in our recent CB paper on *Nasonia*. Indeed, we had found evidence for a diminished role of Toll signaling even in earlier work on the beetle *Tribolium* (65). To show that our new findings for *Oncopeltus* provide an important step in an evolutionary progression we have expanded Figure 1 and included *Tribolium* as well as *Nasonia*. We also added more material to the Introduction and Discussion to explain the difference between *Nasonia* and *Oncopeltus*. In brief: *Nasonia* represents a highly derived system as it is the only insect known so far which establishes the DV pattern in a bipolar manner. The BMP gradient appears to emerge from a maternal source along the dorsal midline independently from Toll and without transport through a ventrally expressed inhibitor. Sog is missing from the genome. In this respect *Nasonia’s* DV patterning system is more derived than that of *Drosophila*. *Nasonia Toll* on the other hand remains responsible for ventral fates providing highly refined patterning information along the ventral midline. The *Toll dpp* double knockdown in *Nasonia* lacks mesoderm, in contrast to the double knockdown in *Oncopeltus*. Taken together while BMP is the dominant morphogen in *Nasonia* both the formation of the BMP gradient and the continued requirement of Toll signaling for the mesoderm/mesectoderm formation are in strong contrast to the DV patterning system of *Oncopeltus* that clearly represents an ancestral state with similarities to spiders and vertebrates.

*1) More information on the expression patterns of the different players in the BMP pathway is essential: Where is* dpp *expressed? It is implied that* dpp *is no longer under Toll-dependent regulation in the milkweed bug (e.g.*
Figure 1
*diagram), however no in situ data is included. The authors also suggest that multiple BMP ligands may be supporting activation of this signaling pathway. Their expression should be shown and discussed. Could they be repressed by Dorsal?*

The expression of the components of the BMP system including *dpp*, a second BMP ligand (*gbb*), *tolloid* and *twisted gastrulation* (*tsg*) are now shown in Figure 2—figure supplement 1. All of these components are either uniformly expressed or have extremely low expression levels, so that they cannot be detected at early blastoderm stages when their activity is required for BMP gradient formation. We know that the probes work since they detect the conserved local expression in late stages. Very low levels are apparently sufficient to support patterning. A similar situation was observed in *Tribolium* and *Nasonia* where early *dpp* and *tsg* are weakly and evenly expressed at the stages when the BMP gradient forms ([65]; Fonseca et al., 2010; Özüak et al., 2014). Thus, regulatory inputs of Toll signaling on these components are alreadylacking in *Tribolium* and *Nasonia* and might have evolved only in the lineage leading to *Drosophila*.

*2) What is the expression pattern of Tolloid? Similar arguments to point 1*.

See response to point 1.

*3) The ability to trace the expression of the Zen homologue as an indicator of maximal BMP signaling in different backgrounds could be very informative*.

Low levels of *Oncopeltus zen* expression are found along the entire circumference of early blastoderm embryos and resolves into a complex pattern that spans the egg circumference at the differentiated blastoderm stage (46). This is one example of many genes dorsally expressed in *Drosophila* and other holometabolous insects, which we tried to use as marker genes and found that they are absent, evenly expressed, or have altered expression domains during blastoderm stages. The failure of the candidate gene approach highlights one of the difficulties connected to working with a hemimetabolous insect. Many aspects of DV patterning are apparently not conserved. In the future we plan unbiased genomic approaches (RNA-seq) combined with RNAi KD to identify new marker genes.

*4) More information on the pattern of the Toll gradient shape would be important*.

The closest approximation of the Toll activation gradient is the nuclear Dorsal gradient since it has not been possible in insects to directly detect activated Toll receptors or the distribution of the Toll ligand Spätzle. We agree that it would be crucial to show the nuclear Dorsal gradient and have raised antibodies against *Oncopeltus* Dorsal1 already at the beginning of our study. However, the antibodies are only useful for western blots and have not worked for whole mount stainings. The fixation methods for early *Oncopeltus* blastoderm embryos are excellent for ISH, but do not reliably preserve protein epitopes. The pMAD stainings provided a lucky exception. We have also tried transient expression essays with GFP constructs. But this was not successful so far. In the future we plan to establish transgenesis and, using the genome which is close to completion, we want to perform genome editing with the CRISPR/Cas9 system.

Given the lack of Dorsal staining we have added cactus in situs. This had been our proxy for Toll activity in *Nasonia* where we also lack anti-Dorsal antibodies. In all insects studied so far including *Drosophila* (55) cactus is an early target gene of Toll signaling and this seems to be an ancestral regulatory circuitry since it has also been observed as an essential element in the innate immune system. *Oncopeltus* possesses several cactus paralogs. Strong knockdown of early expressed paralogs lead to a failure of normal blastoderm formation. However, weak knockdown shows mild ventralization, indicating that at least two of the paralogs are involved in DV patterning. One of them shows early ventral expression which encompasses approximately 60-70% of the egg circumference (the domain is wider than that of *sog*). This indicates that early Toll signaling extends into the dorsal half of the embryo, a prerequisite for explaining the Sander experiment. These considerations are now addressed in the manuscript with the additional *cact* data (new Figure 7).

*5) Additional support for the* Toll1-dpp-*RNAi results should be provided. (one may worry that the Toll1 RNAi support weak activation that becomes apparent in the absence of BMP signaling; What does* Toll1-dl1-dpp *RNAi look like? How about Toll1-tld RNAi?)*.

For the epistasis analysis we have chosen the two genes which give the most complete and most penetrant knockdowns: *Toll1* and *dpp*. All other Toll signalling or BMP signalling components give more variable knockdown results (Figure 2—figure supplement 2). This applies in particular to *dorsal1* as there are two *dorsal* genes (*dl1* and *dl2*) which as single KD lead only to partial dorsalization. Most importantly we are sure that we are dealing with double KD embryos as we observed a phenotypic feature of the *Toll* KD in the background of the *Toll*-*dpp* KD: the anterior shift of *twi*, *sim* and *sog* domains. We have now more explicitly described the function of Toll signalling in AP patterning (Figure 6). We are confident that other double KD combinations like *Toll*-*tld* or *dl1*-*pp* could not provide clearer results.

In this context it is important to note that elevated BMP levels also provide strong evidence for a fundamental difference in ventral gene regulation between *Oncopeltus* and the known Holometabolous insects. In *Drosophila* and *Tribolium*, *twi* and *sog* cannot be repressed by elevating BMP levels. Toll/Dorsal rigidly determines their expression domains. However, in *Oncopeltu*s they are readily repressed by elevated BMP, indicating that their expression state is primarily dependent on the BMP levels and not on Toll/Dorsal.

*In addition, it would be interesting to directly define the shape of the Dorsal gradient (e.g. by indirect immunofluorescence if antibodies are available or could be produced in a reasonable length of time) as this shape is an important output of the modeling. In*
Box 1*, the first diagram suggests that the height and width of the Dorsal gradient increases in time. What is the evidence to support this in the milkweed bug? In*
Box 1*, the second diagram shows that in the middle of the* “*twinned*” *embryo, a peak of BMP signaling abuts a domain of high* sog *expression*. *How is this possible?*

See response to point 4.

*The authors should also discuss how* sog *can be expressed in a small ventral domain in dl1-RNAi mutants. There must be some* “*polarity cue*” *and yet development does not proceed normally. So the levels of Dorsal are important to some degree? Previous studies in* Drosophila *have provided evidence that genes expressed in lateral regions (e.g. brinker) are ubiquitously expressed in* dl dpp *double mutants. Is* sog *expressed ubiquitously within* dl dpp *double* Drosophila *mutants?*

In contrast to the *Toll* knockdown, the *dl1* knockdown does not cause a complete disruption of Toll signalling (possibly due to redundancy with *dl2*). Despite the fact that all *dl1* knockdown embryos have residual *sog* the majority lack *twi* and *sim*. Thus, we have to assume a threshold for *sog* which has to be exceeded to initiated stable patterning.

*Sog* is not expressed in *dl-dpp* double mutants since its activation is strictly dependent on Dorsal like in all other insects studied so far except in *Oncopeltus*. The same applies for early *brinker* expression in *Drosophila*. It is only the late *brinker* (shortly before gastrulation) that is expressed in *dl-dpp* mutant embryos since at this stage *dpp* is required to repress *brk*.

Reviewer 2:

*[…] In summary, this is a very interesting study and the results are significant. However, either additional experimental evidence must be included or the title should be revised. If the study is revised to refocus on the broad role of BMP signaling, then the novelty of the study is perhaps compromised by the authors' recent study in* Nasonia *(Ozuak et al., 2014).*

*Specific comments*:

*1) Where is* dpp *expressed? It is implied that* dpp *is no longer under Toll-dependent regulation in the milkweed bug (e.g.*
Figure 1
*diagram), however no in situ data is included. The authors also suggest that multiple BMP ligands may be supporting activation of this signaling pathway. Their expression should be shown and discussed. Could they be repressed by Dorsal?*

*2) Previous studies in* Drosophila *have provided evidence that genes expressed in lateral regions (e.g. brinker) are ubiquitously expressed in* dl dpp *double mutants. Is* sog *expressed ubiquitously within* dl dpp *double* Drosophila *mutants?*

*3) In*
Box 1*, the first diagram suggests that the height and width of the Dorsal gradient increases in time. What is the evidence to support this in the milkweed bug?*

In Box 1 we assumed that Dorsal (blue) decreases in time. In *Tribolium* the upregulation of *cactus* leads to decreasing nuclear Dorsal concentrations and a disappearance of the gradient. The same is likely to be the case in *Nasonia*. If *cactus* has a similar function in *Oncopeltus* it also would attenuate Toll signalling (see point 4). For the purpose of modelling we want to show in Box 1 that the BMP system is self-organizing and after initiation through Dorsal (by activating *sog*) becomes independent from Dorsal.

*4) In*
Box 1*, the second diagram shows that in the middle of the* “*twinned*” *embryo, a peak of BMP signaling abuts a domain of high* sog *expression. How is this possible?*

We are now explaining the Sander experiment in more detail. Sander used a guillotine to fragment the embryos in two separate halves along the lateral midline. Thus, there is no flow of signalling proteins between the dorsal and ventral fragments. Therefore, high BMP can be exposed to high Sog along the lateral plane bisection.

Reviewer 3:

*[…] In general, the detailed dissection of developmental pathways in organisms that are removed from* Drosophila*, which is successfully used by the Roth lab, and especially the deviations from what we know in* Drosophila*, provide a fresh view not only on the evolution of these pathways, but also on their regulatory logic. This work is a case in point*.

*Several experiments were missing and would provide a broader view of the topic*:

*1) What is the expression profile of* dpp *in* Oncopelus *embryos at different stages? This is important because it would provide the context to interpret the results in Sog RNAi embryos. Is the global activation of the BMP pathway a result of BMP diffusion from the dorsal region, or a reflection of a uniform expression of* dpp*? If* dpp *expression is restricted to the dorsal region, is the repression of expression in the ventral region dependent on Toll signaling, similar to the induction of Sog?*

*2) What is the expression pattern of Tolloid? Similar arguments to point 1*.

*3) What is the nuclear distribution pattern of Dorsal1 and Dorsal2 in wt embryos, and in embryos in which* dpp *signaling has been eliminated?*

*I know that this analysis would require a significant work of raising antibodies, but perhaps it could be circumvented by injection of a plasmid expressing tagged or GFP-linked Dorsal1. It would be important to see the pattern of the Toll activation gradient, and to assess if it extends sufficiently into the dorsal side, to account for the proposed bias in that region in the that is relevant to the Sander experiment (*Box 1*). It would also be important to demonstrate that the Toll gradient is the primary symmetry-breaking event, and is not affected by the loss of the downstream BMP pathway*.

*Several points should be clarified further in the text, without a requirement for additional experiments*:

*Sog levels were not erased but only reduced in the Sog RNAi embryos (*Figure 2*). While the residual Sog is now uniformly expressed, there still appears to be some difference between the level of pMad in the dorsal vs. ventral parts of the embryo (*Figure 3*)*.

Figure 2 shows the embryo from the ventral side. Residual *sog* expression is present only ventrally and correlates with a weak pMAD gradient in *sog* RNAi embryos. Like the residual *sog* expression in *dl1* RNAi embryos this is not sufficient to initiate stable patterning since 100% of the *sog* RNAi embryos lack *twi* expression. Apparently the system shifts only into a stable patterning regime if a certain threshold for pMAD asymmetry is exceeded.

This would not agree with the model, but perhaps the embryos shown are not representative, or maybe there is still some bias in the expression of Sog which resolves only to a very shallow pMad pattern because of the limited levels of Sog?

*The central point of the paper is the switch from a uniform distribution of Sog, to the restricted expression of Sog in the ventral and ventro-lateral domains, and the complementary pattern of BMP activation, triggered by the ventral bias of Toll signaling. The ability to convert even a small bias (as in the middle panel of*
Figure 8*) to a bistable robust BMP pattern, clearly relies on transport of the Sog-BMP complex and its cleavage in the dorsal region by Tolloid. In this respect the logic of modulating BMP signaling is similar to what was described in* Drosophila*. It is not clear however whether this process is also responsible (as it is in* Drosophila*) for generating a graded pattern WITHIN the domain of BMP signaling. As far as I could tell, the pMad levels in embryos that are depleted for Sog were comparable to wt, suggesting that in the absence of inhibition by Sog, the levels of* dpp *are sufficient to elicit maximal signaling and no further concentration of the ligand is required. Also, the late patterns of pMad show sharp borders and fairly uniform expression within the dorsal region (*Figure 3*). I would like to see some more discussion of this issue, regarding the question whether a graded BMP activation pattern is or is not generated within the dorsal region*.

We agree that there is no detectable BMP gradient within the early BMP domain and that the *sog* as well as the *tsg* knockdowns produce uniform high levels of BMP signalling similar to those found in wt within the dorsal 30%. We are now commenting on these observations in the Discussion section. In *Oncopeltus*, a more fine grained pattern of BMP signalling with a sharp dorsal peak is established later during gastrulation and requires the function of the second BMP ligand Gbb as well as the local up-regulation of *dpp* transcription. We believe that during blastoderm stages the DV fate map of *Oncopeltus* does not reach (most notably at the dorsal side) the high spatial precision known from (the long-germ insects) *Drosophila* and *Nasonia.* This precision is only established at later stages. The situation is similar to (the short germ insect) *Tribolium* (Fonseca et al., 2010). We are currently in the process of preparing a manuscript, which addresses these later events.